# Impending anthropogenic threats and protected area prioritization for jaguars in the Brazilian Amazon

Juliano A. Bogoni [1,2 ✉], Valeria Boron[3], Carlos A. Peres [2,4], Maria Eduarda M. S. Coelho[5], Ronaldo G. Morato [6] & Marcelo Oliveira-da-Costa[5]

Jaguars (*Panthera onca*) exert critical top-down control over large vertebrates across the Neotropics. Yet, this iconic species have been declining due to multiple threats, such as habitat loss and hunting, which are rapidly increasing across the New World tropics. Based on geospatial layers, we extracted socio-environmental variables for 447 protected areas across the Brazilian Amazon to identify those that merit short-term high-priority efforts to maximize jaguar persistence. Data were analyzed using descriptive statistics and comparisons of measures of central tendency. Our results reveal that areas containing the largest jaguar densities and the largest estimated population sizes are precisely among those confronting most anthropogenic threats. Jaguars are threatened in the world's largest tropical forest biome by deforestation associated with anthropogenic fires, and the subsequent establishment of pastures. By contrasting the highest threats with the highest jaguar population sizes in a bivariate plot, we provide a shortlist of the top-10 protected areas that should be prioritized for immediate jaguar conservation efforts and 74 for short-term action. Many of these are located at the deforestation frontier or in important boundaries with neighboring countries (e.g., Peruvian, Colombian and Venezuelan Amazon). The predicament of a safe future for jaguars can only be ensured if protected areas persist and resist downgrading and downsizing due to both external anthropogenic threats and geopolitical pressures (e.g., infrastructure development and frail law enforcement).

[1] Universidade de São Paulo (USP), Escola Superior de Agricultura "Luiz de Queiroz" (ESALQ), Departamento de Ciências Florestais, Laboratório de Ecologia, Manejo e Conservação de Fauna Silvestre (LEMaC), Piracicaba, SP 13418-900, Brazil. [2] School of Environmental Sciences, University of East Anglia, Norwich, Norwich NR4 7TJ, UK. [3] WWF-UK, The Living Planet Centre, Rufford House, Brewery Road, Woking, Surrey GU21 4LL, UK. [4] Instituto Juruá, Rua Belo Horizonte 19, Manaus, Brazil. [5] WWF-Brasil, CLS 114 Bloco D-35, Brasília 70377-540, Brazil. [6] Centro Nacional de Pesquisa e Conservação de Mamíferos Carnívoros, Instituto Chico Mendes de Conservação da Biodiversidade, Estrada Municipal Hisaichi Takebayashi 8600, Atibaia 12952-011, Brazil. ✉email: bogoni@usp.br

Large terrestrial carnivores such as jaguars Panthera onca exert critical roles in maintaining ecosystem health and integrity[1,2]. However, many populations are rapidly declining, and are particularly vulnerable to local extinction because the species occurs at low densities, experiences slow population growth rates, and requires large areas containing a healthy prey base to survive[3–6]. Thus, their long-term population viability requires large-scale conservation planning approaches that include networks of protected areas and connectivity corridors (e.g., refs. [7,8]). Virtually all large-bodied wild carnivore species have experienced population declines worldwide[2,5]. These apex predators are markedly susceptible to high mortality in areas densely populated by humans[5,9], despite often tolerating agroecosystems as either corridors or supplementary habitats in fragmented landscapes[10–12].

The jaguar is the world's third largest extant felid and the largest in the Americas[13]. As other apex predators, jaguars exert considerable top-down control on vertebrate populations. They have populated the imagination of people since pre-Columbian days. The jaguar is therefore considered an emblematic flagship and a keystone species[1,14]. Due to their large spatial requirements, jaguars are also considered an umbrella species[15,16] and are valuable in conservation planning, ensuring that many other co-occurring species and high-quality habitats are protected[15]. For instance, the Jaguar 2030 Roadmap, a range-wide plan to conserve jaguars in priority landscapes and corridors would additionally benefit a suite of co-occurring vertebrates[17]. The species ranges from the southern USA[18] to Argentina and is considered "Near Threatened" according to the IUCN Red List[19], and "Vulnerable" in Brazil[17]. Jaguars prey on a broad range of large-bodied terrestrial, semi-aquatic, and aquatic prey[18,20], resulting in large spatial requirements and wide-ranging movements to meet their daily metabolic needs[19,21]. Home range sizes tend to increase as habitat quality decreases, rendering these apex hyper-carnivores particularly vulnerable to habitat loss and fragmentation[10,12].

Despite the cultural and ecological importance of the iconic jaguar, the species only occupies ca. 50% of its historic range[22] and has been almost extirpated from heavily modified Brazilian biomes, such as the Atlantic Forest and the Caatinga[23]. The main threats to jaguar survival are habitat loss, human persecution, and decline of prey populations[22]. The Amazon forest still holds large numbers of jaguars and ~67% of the entire contemporary range of this species (~9 million km²), where the jaguar has the highest probability of survival[4,24,25]. Forests across the Brazilian Amazon comprise ~77% of the Pan-Amazon region of South America[26], making it a high-priority stronghold for jaguar conservation.

Despite a large network of protected areas (hereafter, PAs), the Brazilian Amazon has been encroached by deforestation frontier expansion, driven by unnatural (i.e., human caused) wildfires, agriculture and cattle ranching, mining, and roads[27,28], making conservation priority-setting actions increasingly necessary[29–31]. Amazonian deforestation rates have recently accelerated, leading to a process of savannization of both fauna and flora throughout the so-called "deforestation arc" of the Brazilian Amazon[32,33]. Annual deforestation in the Brazilian Amazon in 2018–2019, estimated at ~1,760,000 hectares, was further aggravated by unprecedented anthropogenic fire events[33,34], with the peak deforestation year in a decade recorded in 2020[34]. Under this complex socio-environmental context, the PA network across the Brazilian Amazon is crucial for jaguars and biodiversity conservation[35,36]. Considering all PAs in the Brazilian Amazon, there are 307 federal and state-managed conservation units (UCs; abbreviation in Portuguese), 196 of which are sustainable use and 111 are strictly protected reserves, comprising 23.5% (~1.18 million km²) of Brazilian Amazonia. An additional 23% of the Brazilian Amazon (~1.16 million km²) is protected 'on paper' by 424 indigenous reserves (IRs), but their fate remains highly uncertain against the human-induced pressures (e.g., mining and persecutions), such as legal challenges and invasions[37].

Tropical forest reserves are recognized as pivotal tools in retaining relatively intact biotas worldwide, and buffering tropical forest climate tipping-points by retaining aboveground carbon storage[38]. Considering the vast but severely underfunded network of PAs, the dilemma of prioritizing conservation investments in the short, medium, and long term is paramount for successful conservation outcomes[39]. A fine-tuned conservation plan for a focal species such as the jaguar can serve as a robust proxy for overall biodiversity persistence[15]. An analogous study evaluated the performance of PAs in maintaining viable populations of African lions (Panthera leo) and their prey[5]. This range-wide assessment of PAs revealed multiple socio-environmental threat factors, such as hunting, human and livestock encroachment, and human-wildlife conflicts[5]. However, PAs within the range of lions exert the cumulative potential to host a large global population size. However, the effectiveness of a PA network depends upon the legal status, security, and particularly the management and enforcement actions[5]. Across the Neotropics, jaguar population declines also coincide with similar human-induced pressures (see ref. [12]). PAs are central to safeguarding biodiversity, yet these protected lands are under multiple geopolitical pressures and their nominal buffer zones are typically as degraded as the wider unprotected countryside[40].

Here, we (1) identify and quantify the main socio-environmental threats to jaguars across the Brazilian Amazon PAs, (2) assess to what degree these threats are related with jaguar population sizes within PAs, (3) assess whether the legal denomination of PAs affects jaguar population sizes and threats, and (4) identify PAs that merit high-priority short-term conservation action to safeguard these apex carnivores (i.e., PAs with high threat levels and hosting large jaguar populations). Our hypotheses are that (1) habitat degradation factors, such as deforestation and wildfires, are the most important and imminent threats to jaguar survival across the Brazilian Amazon; (2) the legal denomination of PAs is an important determinant of PA threat status; and (3) PAs safeguarding large jaguar populations, which should be prioritized for jaguar conservation, are precisely those confronting the most severe habitat degradation threats. Our study is timely to better understand current threats to jaguars, and inform conservation planning by presenting an evidence-based agenda for jaguar conservation in the Amazon.

## Results

**Low bias of jaguar densities used to estimate PAs jaguar numbers and buffer size evaluation.** Mean jaguar density inside PAs and 5 km buffers across the Brazilian Amazon was 2.06 (±0.9 SD) and 1.99 (±0.8 SD) ind/100 km² (Fig. 1A), respectively ($N_{total\ pixels}$ = 128,087). By contrasting all available jaguar density estimates at sites across the species range ($N = 50$) with those predicted by Jędrzejewski et al.[25], we confirm that the latter estimates are both reliable and conservative. The highest population density predicted by Jędrzejewski et al. (2018)[25] was 4.86 (±0.05 SE) per 100 km², whereas the 50 jaguar available density estimates averaged 5.49 (±4 SD; range = 0.51–18.29) individuals per 100 km². Furthermore, 66% ($N = 33$) of all existing jaguar density estimates fall within the Jędrzejewski et al.[25] range of estimates (min-max) for Amazonian PAs (Fig. 1B). We additionally compared recently obtained (not included in ref. [25]) jaguar densities from 13 sites across the Brazilian Amazon—and near the country border (e.g., Peruvian Amazon)—vs. the density estimate values derived from ref. [25] at the same coordinates,

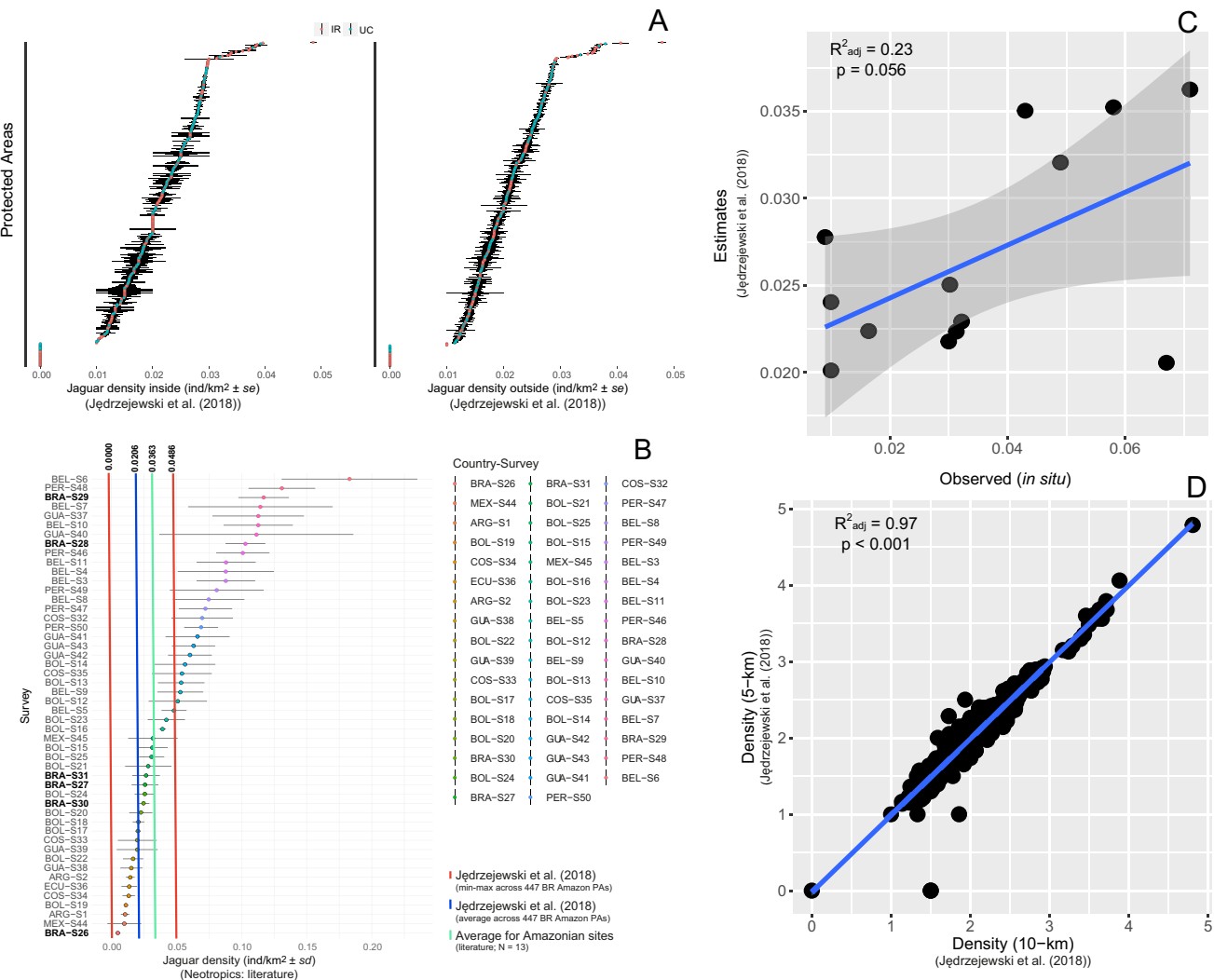

**Fig. 1 Jaguar density bias and buffer size evaluation. A** Jaguar density estimates (derived from ref. [25]) and standard errors (SE) for 447 protected areas across Brazilian Amazon. **B** Summary of field surveys deriving in situ jaguar density estimates and the main trends derived from our bias evaluation of the Jędrzejewski et al.[25] estimates; **C** Jaguar density in situ vs. Jędrzejewski et al.[25] estimates for sites across the Brazilian Amazon (including country-border sites); and **D** Jaguar density estimates within 5 and 10 km buffer areas around PAs across the Brazilian Amazon.

yielding a linear and marginally significant relationship ($R^2_{adj}$ = 0.23; $p = 0.056$) with only one estimate as an outlier (Fig. 1C). Thus, Jędrzejewski et al.[25] on average underestimated jaguar densities by $-0.86$ ($\pm 1.9$) ind/km². Moreover, our buffer size of 5 km was linearly compatible in terms of jaguar densities in relation to a 10 km buffer area ($R^2_{adj}$ = 0.97; $p < 0.001$; Fig. 1D).

**Threats to jaguar across Amazonian protected areas.** Considering the average of 2.06 individuals per 100 km² across the 447 protected areas in the Brazilian Amazon and their respective 5 km radial buffers (hereafter, the wider PA area), this amounted to a combined area of 2,244,090 km², which could support 47,942 (95% CIs: 38,129–57,752) jaguars. Between 2016 and 2019, deforestation across these protected areas and their respective 5 km radial buffers amounted to 5560 km², representing 0.25% of the total area. There were also 101,804 agricultural and forest understory fires over a 5-year period, and roads across these protected areas and their buffers amounted to 3947 km. We found significant differences in both jaguar population sizes and levels of threat among protected area types (Fig. 1; Supplementary Data 2). The largest jaguar populations were concentrated in strictly protected (SPA) and sustainable-use reserves (SUR)

compared to Indigenous Reserves, while the threat index was higher in SURs than in SPAs and Indigenous Reserves (IR1 and IR2; Fig. 2; Fig. 3A).

**Protected area prioritization.** Our proposed threat index was on average TI = 0.05 (±0.09). Higher threat values were concentrated in PAs located within or near agricultural and logging frontiers in the southern Amazon or physically accessible areas (e.g., via rivers; Fig. 3B). Based on the largest threat index contrasted with the largest conservative estimate of jaguar population size of each PA (Fig. 3A), we identified the top-10 non-redundant protected areas (2.24%) that deserve urgent high-priority conservation attention (Fig. 3C; Fig. 4A; Table 1). These 10 PAs amount to a total of 25,254 km² (1.5% of the overall PAs acreage). Under an average threat index of TI = 0.41 (±0.12). They can support a conservative estimated 3,511 jaguars—13.2% of 26,680 total under our conservative approach to estimate jaguar population size for prioritization purposes (which represent –29.5% of uncorrected estimates inside PAs ($N = 37,874$)). Yet these conservative vs. uncorrected estimates of jaguar population size inside PAs were highly correlated (see Supplementary Fig. 2).

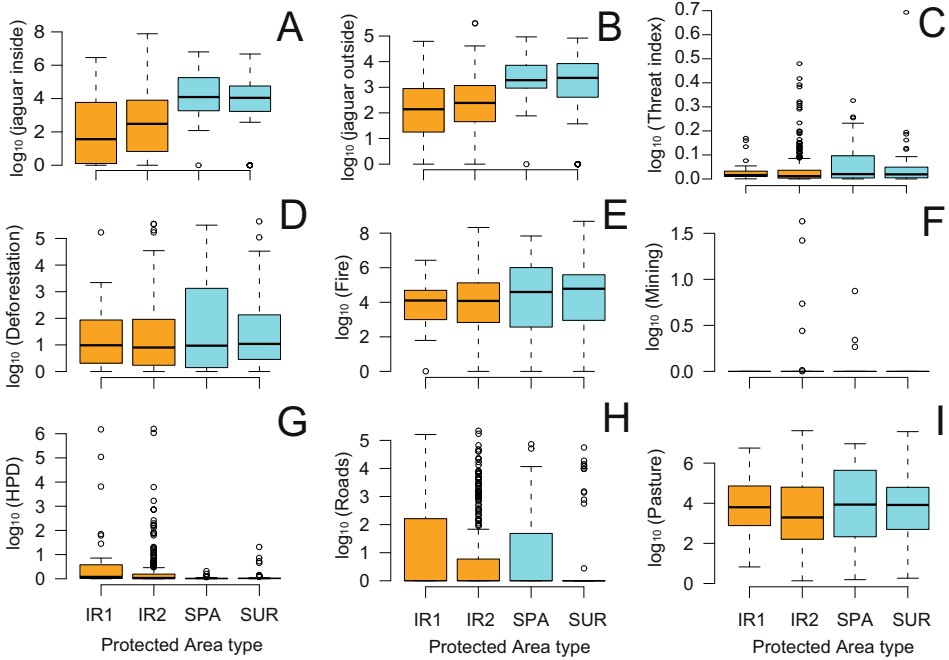

**Fig. 2 Threats to jaguar across Amazonian protected areas.** Jaguar population sizes inside protected areas (**A**), jaguar population sizes outside protected areas (5 km buffer; **B**), threat index (TI; **C**), deforestation (km$^2$; **D**), fire hotspots (N; **E**), mining areas inside (km$^2$; **F**), human population density (persons/km$^2$; **G**), roads (km, **H**), and exotic cattle pastures (km$^2$, **I**) across 447 protected areas grouped by legal status across the Brazilian Amazon. SPA Strictly Protected Conservation Units, SUR Sustainable Use Conservation Units (SUR), IR1 declared Indigenous Reserves, IR2 Indigenous Reserves that have been demarcated, approved, or legally sanctioned. All data are log$_{10}$-transformed and from C to H are the sum of values both inside and outside protected areas.

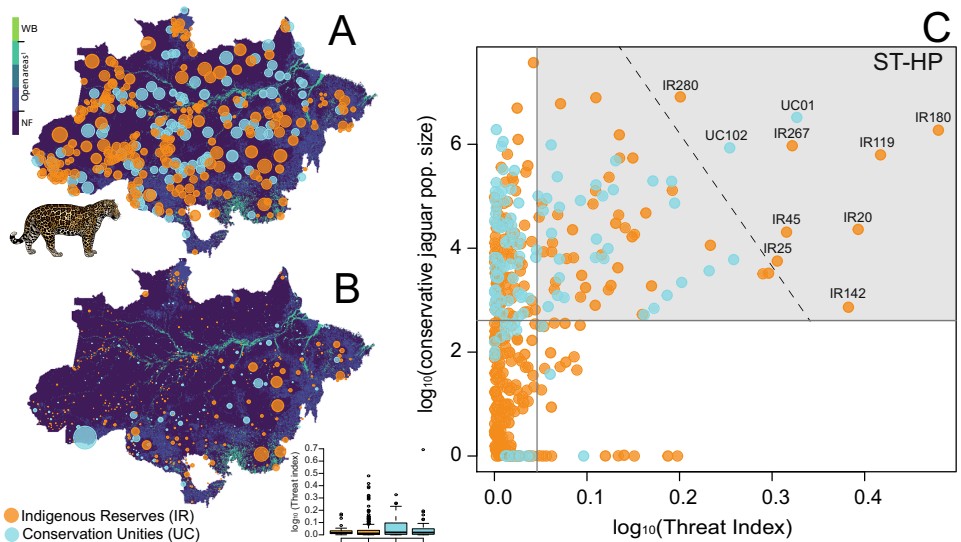

**Fig. 3 Jaguar population size, threat index (TI) and prioritization diagram for jaguar conservation across the Brazilian Amazon.** Distribution of jaguar population size (log$_{10}$ x + 1) inside protected areas (**A**), threat index (TI) and values per protected area type (**B**) and bivariate plot between the threat index and conservative jaguar population sizes inside 447 protected areas across the Brazilian Amazon (**C**). Acronyms are ST-HP: short-term high-priority quadrant (delimited by highlighted gray-frame) and the respective top-ranking 10 areas that should be prioritized in each approach based on the extreme of distribution thresholds by a tangential line. We also identified additional Amazonian PAs that should be prioritized for jaguar conservation in the short to medium term according to our prioritization quadrants (highlighted gray-frames; see Supplementary Data 3). The background map in (**A**) and (**B**) was produced by the authors and represents the land cover and land use dated from 2019[98]. WB water bodies, NF natural forest. ¹Open areas include non-forest natural vegetation, farming, and non-vegetation areas. For all land cover and land-use classes, see ref. [98]. Fernanda D. Abra kindly provided jaguar drawing used in this figure. All other elements in the figure were created by the authors using R code.

These 10 PAs and their respective buffers experienced cumulative deforestation over 4 years of 1175 km$^2$ (20.8% of the total amount of deforestation in all wider PAs, amounting to 5560 km$^2$), 20,941 cumulative fire hotspots over 5 years (20.6% of all 101,804 fires), contain 269.4 km of roads (6.9% of the total road network within PAs and buffer zones of 3,947 km), 11,179 km$^2$ of pastures (23.3% of 47,936 km$^2$ of all pastures across PAs), and an average HPD of 0.009 (± 0.012) per km$^2$ compared to an average of 2.14 (±27.46) for all PAs. These areas —comprising eight indigenous reserves and two conservation

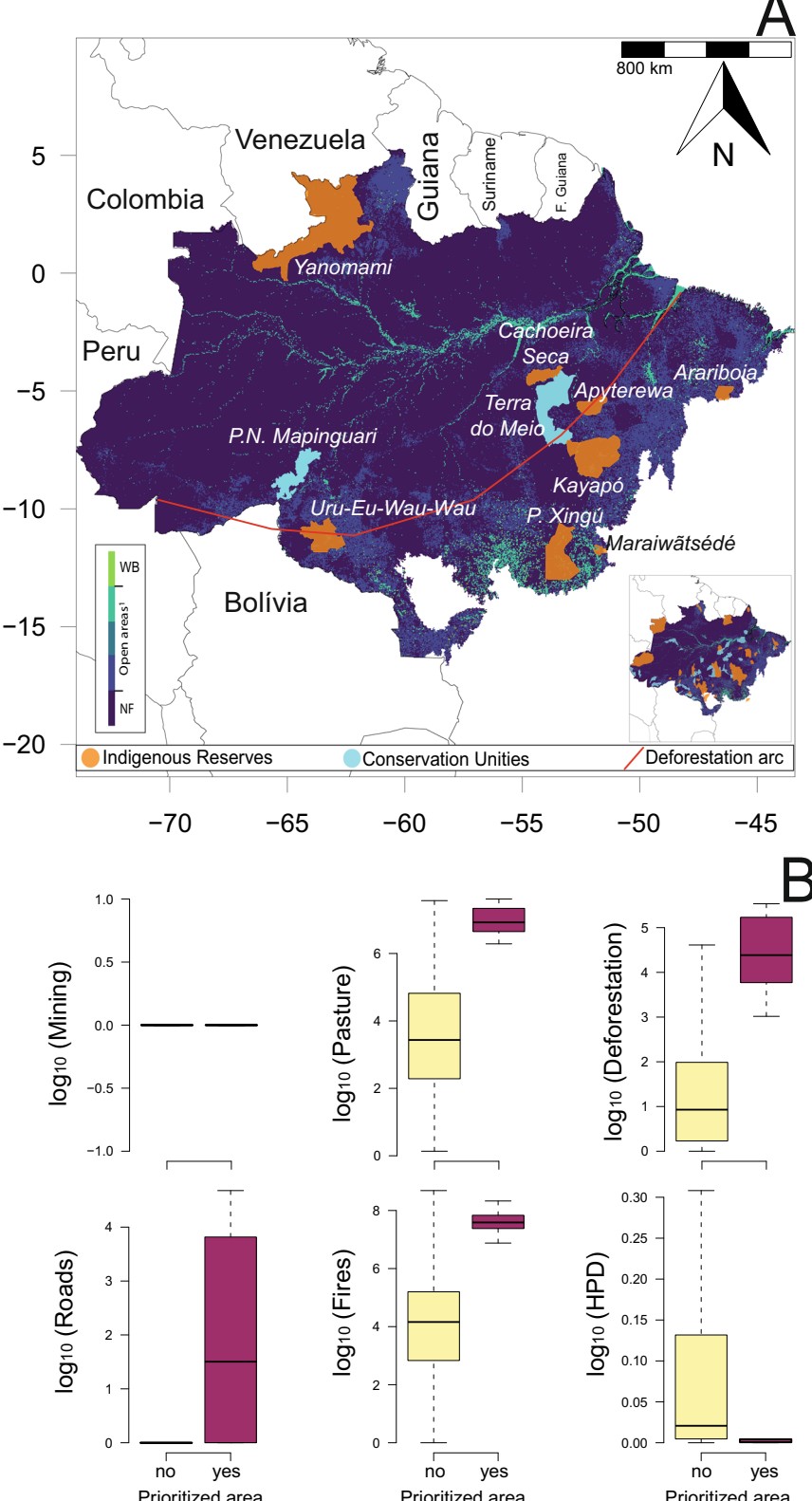

**Fig. 4 Top-10 protected areas in the Brazilian Amazon prioritized for jaguar conservation. A** Location of 10 protected areas in the Brazilian Amazon prioritized for jaguar conservation considering the threat index, jaguar density, and jaguar population size. The background map was produced by the authors and represents the land cover and land use dated from 2019[98]. WB water bodies, NF natural forest. [1]Open areas include non-forest natural vegetation, farming, and non-vegetation areas. For all land cover and land-use classes see ref. [98]. The small map bottom-right represent other 74 PAs that deserves prioritization (see Fig. 3C; Supplementary Data 3); **B** Mining (km²); Pastures (km²); Deforestation (km²); Roads (km); Fire hotspots (N hotspots) and Human population density (HPD) comparison between areas that should be prioritized for jaguar conservation across the Brazilian Amazon. Data in (**B**) are log₁₀-transformed and represents the data inside PAs. All elements in the figure were created by the authors using R code.

**Table 1 Protected area codes, names, size (km²) and legal status of high-priority reserves (i.e., top-10) for jaguar conservation across the Brazilian Amazon (see also Fig. 4 and Supplementary Data 3).**

| Code | Protected area name | Area (km²) | Status | Conservative jaguar pop. size | TI |
|------|---------------------|------------|--------|-------------------------------|-----|
| IR20 | *Apyterewa* | 7740.6 | IR2 | 77.4 | 0.48 |
| IR25 | *Arariboia* | 4161.4 | IR2 | 41.6 | 0.36 |
| IR45 | *Cachoeira Seca* | 7341.4 | IR2 | 73.4 | 0.37 |
| IR119 | *Kayapó* | 32,807.4 | IR2 | 328.1 | 0.52 |
| IR142 | *Maraiwãtsédé* | 1652.4 | IR2 | 16.5 | 0.47 |
| IR180 | *Parque do Xingu* | 26,413.7 | IR2 | 528.3 | 0.62 |
| IR267 | *Uru-Eu-Wau-Wau* | 19,555.3 | IR2 | 391.1 | 0.38 |
| IR280 | *Yanomami* | 100,303 | IR2 | 1003 | 0.22 |
| UC01 | *Estação Ecológica da Terra do Meio* | 33,779.4 | SPA | 675.6 | 0.39 |
| UC102 | *Parque Nacional Mapinguari* | 18,782.8 | SPA | 375.7 | 0.29 |

See the prioritization approach in Fig. 3C.
*SPA* Strictly Protected Conservation Units, *IR2* Indigenous Reserves that were delimited, approved or ratified, *TI* threat index.

units—are located near the Amazonian 'arc of deforestation' and the northern Amazon (Fig. 4A; Table 1). These 10 PAs exhibited significantly higher deforestation rates (F = 62.7; $p < 0.001$), more severe or more frequent fires (F = 42.2; $p < 0.001$), larger pasture areas (F = 40.5; $p < 0.001$), and the largest road networks (F = 6.4; $p = 0.01$), but did not differ in terms of mining (F = 0.1; $p = 0.73$) and HPD (F = 1.2; $p = 0.27$) compared to other protected areas (Fig. 4B).

We also identified 74 (16.6%) additional Amazonian PAs (41 IRs and 33 UCs) that should be prioritized for jaguar conservation in the short- to medium term according to our prioritization quadrant (Fig. 3C; Fig. 4A; Supplementary Data 3), given that these PAs had a higher threat index and larger conservative jaguar population size than the average for all PAs studied. These additional PAs encompass 666,723 km² (38.0%) and could support and additional 10,650 jaguars (40.0%) (Fig. 4A; Supplementary Data 3). Thus, within only 84 protected areas we could protect in the short-term 53.1% (N = 14,161) of the entire conservative jaguar population size (N = 26,680) estimated to occur within all 447 PAs across the Brazilian Amazon. This amounted to ~19% of all 447 PAs across the Brazilian Amazon and encompassed 39.0% of the overall protected acreage.

## Discussion

The persistence of healthy jaguar populations across their extensive Amazonian stronghold depends on the enforcement of public policies and legislation that safeguard the network of protected areas, indigenous territories, and their respective buffer zones. Since jaguars are classed as a flagship, umbrella[15] and a keystone species[41], the entire Amazonian biota can benefit from conservation efforts focused on this large felid. Our results reveal that the areas containing the highest jaguar population densities and largest estimated population sizes are precisely among those most pressured by anthropogenic impacts in terms of habitat degradation. Deforestation, agricultural expansion including cattle pastures and cropland, and wildfires are prevalent in protected areas hosting the largest estimated jaguar populations, especially within their buffer zones, which fare far worse than their adjacent PAs[40]. As a large-bodied apex-predator, the large home range requirements of jaguars[19] frequently expose them to the edges and buffer zones of protected areas, coinciding with sites experiencing the most severe levels of habitat degradation. This contagious "edge effect" could determine jaguar numbers inside protected areas, by increasing mortality rates through, for example, shootings and roadkills, and the perverse effects of habitat fragmentation created by deforestation[42,43].

With more than 20 transboundary jaguar populations across the range[44,45], our study identified key PAs for jaguar conservation, some of which are located in transboundary regions that require immediate action. Across the Neotropics, drivers of local biotic depletion have accelerated since the 1970s. Dominant anthropogenic disturbances that lead to species declines and local extinctions include access to hitherto isolated forested areas via new roads[46], wildfires fueled by climate change[42,43], deforestation due to agribusiness frontier expansion[44,45], relaxation of environmental law enforcement[47], increasing hunting pressure[48], and the synergistic combinations between these and other socioeconomic stressors[49]. The Brazilian Amazon experienced a multifaceted spike in environmental degradation over the last decade, exacerbated by a renewed acceleration in deforestation and human-induced fires[34], exerting further pressure on Amazonian forest wildlife and native ethnic groups[37,50].

Our results revealed that some areas that are important for jaguars (large jaguar population sizes) are experiencing high fire severity (see Fig. 4B). Typically, fire events represent the "coup de grace" following forest degradation, particularly where soil hydrological deficits are exacerbated. As powerful drivers of habitat degradation, deforestation and fires are historically synergistic, exerting a double-negative effect on tropical forest biotas[51]. Amazonian surface wildfires trigger a cascade of detrimental effects on biodiversity, particularly in areas that experienced little or no fire-stress over evolutionary timescales, leading to wholesale changes in species turnover[52]. It is therefore concerning that fire events are concentrated in areas containing large populations of jaguars and many other vertebrates. Our analysis also shows the potential risk of livestock pastures, which had also proliferated in areas packing large jaguar populations. Cattle ranches in the Amazon have two important negative impacts on jaguars. First, exotic pastures directly result in habitat loss for forest wildlife and severe impacts on biodiversity[49]. Secondly, pasture-dominated landscapes become demographic sinks for jaguars, where as many as 110–150 large felids can be killed annually within a single Amazonian county through poisoned carcasses and direct persecution by professional jaguar/puma hunters and ranch staff[53].

Previous studies in human-modified landscapes show that habitat loss and fragmentation have a strong detrimental impact on jaguar populations, which are now locally extinct in several Neotropical ecoregions[23,49]. For instance, the few remaining jaguar subpopulations in the Atlantic Forest are small, highly dispersed, and highly isolated within a few sufficiently large forest remnants[23,54]. Our evidence indicates that consolidated Amazonian deforestation frontiers could exhibit a similar negative spiral for jaguar demography within a few decades. For instance, jaguar populations declined by 1.8% in the last 5 years due to deforestation and wildfires[55]. Jaguars have large spatial requirements

and home range sizes, so population density depends on high-quality habitat providing an ample prey base[12,19]. In addition, jaguars strongly avoid non-forest areas embedded in highly-forested landscapes[56]. Deforestation can lead to further environmental degradation, including mining, roads and overhunting[57], increasing the threats to jaguars across their largest forest stronghold.

An additional factor of direct jaguar mortality is the expanding road network, particularly within areas surrounding PAs. Although the road network across the Amazon is still incipient, new major road projects can rapidly change this scenario and result in roadkills, and further habitat degradation, deforestation, and access by hunters to previously remote areas[58,59]. The ubiquitous presence of road networks causes negative effects on mammal populations up to 5 km[60], strongly affecting apex predators across the tropics (e.g., ref. [61]). Mitigation via underpasses has shown positive results in protecting vertebrates from roadkill[62]. In other Neotropical biomes, such as the Atlantic Forest and the Cerrado, roadkills are an important contributor to jaguar mortality, further removing individuals from already depleted populations[23]. New government plans to expand the road network across the Amazon are an additional threat for jaguars. Thus, rethinking the strategic deployment of new infrastructure both for people and the environment is critical[59].

These threat patterns are common to other apex predators (e.g., ref. [5], ref. [63]). For instance, tiger populations (*Panthera tigris*) are highly threatened by recurrent forest loss across Indochina[63]. Currently, tigers only survive in forest ecosystems, and core breeding populations are restricted to protected areas across much of their original range[63]. Similarly to tigers and other wide-ranging large carnivores, a comprehensive network of effective protected areas throughout the Amazon is key for the persistence of jaguar populations (e.g., ref. [8], ref. [61]).

The largest jaguar populations were in strictly protected conservation units (SPAs), followed by multiple-use reserves (SURs) and legally demarcated and sanctioned Indigenous Reserves (IRs). However, SURs were exposed to significantly higher levels of threat than SPAs and IRs (higher cumulative threat index), even if individual threat variables comprising our threat index did not vary between reserve denomination types (see Fig. 1 and Supplementary Data 2). The 4-year deforestation time-series reached 0.25% (5560 km$^2$) of the combined acreage across all protected areas and their respective buffers, an area 3.7-fold the size of São Paulo, the largest Latin American city. Moreover, cumulative fires over 5 years in the overall PA area exceeded 100,000 burn hotspots. Other threats such as mining, road expansion, cattle pastures, and growing non-indigenous populations are also increasing across the Brazilian Amazon due to greater agricultural investments and dismantling of environmental legislation and enforcement. This also shows that, through sheer lack of conservation investments, Amazonian nature reserves have begun to fail their biodiversity conservation mission statements, much like early models of reserve defensibility predicted[64], and this predicament is often worse for indigenous territories. This study is also a warning to policy-makers as the situation will likely become worse in the near future, unless Amazonian PAs can be effectively protected.

Brazil hosts over 50% of the global jaguar population[23] and is a signatory of the Jaguar 2030 Roadmap and the Convention of Migratory Species (CMS)—which includes the Jaguar[44,45]. However, these commitments have not turned into implementation with ever decreasing funding for PAs. The country's state-managed protected areas are grounded in strict legislation under the National System of Conservation Units (see Sistema Nacional de Unidades de Conservação (SNUC; 2000)[65]), but over the past few years, elevated geopolitical pressures have greatly

weakened the management capacity of these areas. Indigenous Reserves (also known as Indian Lands) are officially recognized to secure territorial rights for indigenous peoples and their traditional cultures, but withhold no legal property ownership (whether private or communal) over their own lands given that they are still demarcated as public lands[37]. Across the 447 protected areas examined here, indigenous reserves were predicted to contain ~24,000 jaguars, representing 63.2% of the total estimated number of jaguars across the Brazilian Amazon's ~224 million hectares of protected areas. The importance of indigenous reserves is intrinsically linked to their larger sizes and larger wildlife populations compared to most state reserves, while ensuring legitimate land claims for native Amazonians as their original stewards and landholders[66].

Conservation priority-setting exercises are highly context-dependent in terms of socioeconomic dimensions[31] and prioritization dilemmas apply to poorly implemented protected areas that deserve urgent attention. Conservation efforts are typically limited by financial resources and economic models that assign priorities to these efforts have gained increasing importance[67]. We showed that our proposed threat metric could be an important priority-setting tool, and we were able to identify at least 10 top-ranking PAs that deserve immediate conservation efforts due to their large jaguar population sizes and high threat levels, and an additional 74 PAs that should be prioritized in the short-medium term (see Fig. 4A; Supplementary Data 3). Interestingly, this approach identified that the geographic distribution of this set of protected areas is highly congruent with the Amazonian 'arc of deforestation'. This region hosts the world's largest mechanized agriculture frontier and includes the transitional ecotone between Amazonian forests and the Cerrado wooded scrublands, encompassing the Brazilian states of Maranhão, Tocantins, Pará, Mato Grosso, Rondônia, Acre, and more recently Amazonas. Chronic deforestation across the vast deforestation arc facilitates human-wildlife conflicts between landowners and large felids, within which the latter usually loses[53]. Moreover, climate models showed that large parts of the Amazon affected by the 'deforestation arc' might reach a tipping point at which they will be transformed into savannahs[68], representing a further challenge for jaguar conservation. This also calls for a more detailed analysis on the source-sink demographics of large cats in increasingly deforested hyper-fragmented landscapes that typically set reproductive viability thresholds for apex carnivore populations, such as the Harpy Eagle[69].

Our quadrant of short-term prioritization (Fig. 3C) revealed that the vast majority of PAs (including the top-10) were located in the frontline of the deforestation arc and across transnational frontiers, mainly with Bolívia, Peru, Colombia, Venezuela, and Suriname. Similar to the deforestation arc, these national frontiers also face environmental and governance challenges. The rapid increase of deforestation and fires, expansion of agribusiness, and the reshaping of the South American geopolitical landscape[70] reinforces the mutual challenge of these aforementioned nations (including Brazil) to preserve its biodiversity and its indigenous peoples through transnational policies and strengthening networks of protected areas and ecological corridors, including partnerships aimed at thinking about the dilemmas of decision-making and prioritizing short-term and long-term conservation investments.

We recognize that our threat index can include decision-making biases, which is typically a compromise in selecting between a wide range of management strategies based on uncertainty and incomplete information[71]. Nevertheless, the 10 protected areas selected for immediate action in the Brazilian Amazon share a common mix of conservation challenges, including significant human population pressures and elevated

levels of deforestation and wildfires. Further, we find significant differences in the main human-induced drivers of habitat degradation between protected areas that should be prioritized compared to PAs elsewhere. Compared to other PAs, these 10 protected areas hosted a 10-fold higher deforestation rate, larger pasture areas and more fires, and are typically geographically positioned in the most pressured regions of the Amazon, including important frontiers regions. However, these top-ranking 10 PAs still host small human populations. Human population density is widely used to anchor geographic gradients of human disturbance and is a powerful proxy of human threats on natural ecosystems, including accessibility, infrastructure, land-use change, and direct mortality[7,72].

We also acknowledge possible bias in the jaguar population size estimates based on ref. [25], such as the buffer size around PA boundaries. Yet, we used data from the literature and variance measures to dissect part of this potential uncertainty. By using the more conservative value of jaguar density across PAs and them categorizing them into density classes, we were able to reduce the bias of the estimates. Moreover, the jaguar density estimates were >70% similar compared to density estimates at jaguar study sites across the Neotropics (i.e., 36 in 50 studies have jaguar density between the min and max of Jędrzejewski et al.[25]; Fig. 1B). Thus, we erred on the conservative side in defining the top-10 PAs that deserves urgent action to ensure jaguar persistence.

Among these 10 PAs, across the Northwest Amazon (near Colombia and Venezuela), for instance, the *Yanomami* (top-10) and *Alto Rio Negro* (top-74) Indigenous Reserves—encompassing 183,323 km² of nearly intact forest and harboring several ethnics groups ($N = 31$; including isolated peoples), such as *Arapaso*, *Mirity-tapuya*, *Yanomami*, and *Ye'kwana* likely contain two of the largest estimated jaguar populations (1003 and 880 individuals, respectively) of the 477 protected areas examined here. At the same time, these indigenous reserves face a threat index of 0.22 and 0.08, respectively, fueled by deforestation, gold mining, timber extraction, fires, and growing human populations, and circa than double the average TI value across all PAs. Other PAs identified at the top-10, such as *Kayapó, Parque do Xingu, Uru-Eu-Wau-Wau, Estação Ecológica da Terra do Meio*, and *Parque Nacional Mapinguari* harbor at least 27 ethnics groups. These PAs account for only 1.5% of the overall PA acreage but concentrate 13.2% of the estimated jaguar population size across Brazilian Amazonia; yet they are facing mounting pressure from agricultural frontiers and mining expansion[32,73].

Other PAs identified at the top-74 (short-term quadrant), such as *Vale do Javari*, harboring the *Kulina Páno, Matis, Matsés* ethnic groups—contain the largest estimated conservation jaguar population (1940 individuals) of the 477 protected areas examined here, but is under a moderate threat value (TI = 0.24) as territorial conflicts and persecution are intensified. We also can highlight the 9761 km² *Parque Nacional da Serra do Divisor* near the Peruvian border. There has been heavy recent political pressure to build a road through this park and link soybean production areas in Brazil with the Pacific. Thus, the short-term priority quadrant ($N = 84$; ~19%), could protect at short-term over 50% of the entire conservative jaguar population size estimated to occur inside PAs in the Brazilian Amazon, and consequently the majestic Amazonian biodiversity.

The vast majority of Brazilian PAs, particularly in the Amazon, confront an average financial insufficiency of 89.7%[74]. In fact, this severe underfunding, understaffing and lack of operational infrastructure has existed since the early 1990s[64] despite several billions of conservation dollars flowing into Brazil to boost the effectiveness of Amazonian PAs. Currently, the Brazilian government invests less than one dollar per km² across all protected areas under state and federal jurisdiction. This is even more

pronounced in the buffer zones surrounding conservation units[40], exacerbating jaguar population declines and weakening connectivity. Moreover, we highlight that the lower-medium-priority 363 protected areas considered here do not necessarily deserve fewer conservation investments. For instance, 84 (19%) of all Amazonian PAs were classified as short-term priority and face above-average levels of threat and hold above-average jaguar densities. We further reinforce that the jaguar density estimate predictions that we used[25] do not include prey abundance data and can be biased. Thus, we recommend that future estimates of large felid population densities should include a measure of prey productivity. Overall, we reinforce that the threats faced by jaguars across the Amazon consistently match the broad geographic patterns of habitat degradation.

The initial hypotheses we posed here were corroborated. Leading drivers of habitat degradation (i.e., deforestation and fires) are impending threats to jaguars as they occur within and around areas containing large numbers of jaguars across the southern and eastern Amazon. Burgeoning human populations along an expanding agricultural frontier will likely inflict further mortality on both resident and transient individuals. Further, other leading drivers such as agricultural conversion continue to threaten PAs that harbor large jaguar population sizes. The most legally secure protected areas under the most restrictive use host higher jaguar abundance, and reserves that should be prioritized for jaguar conservation are located within or near recent deforestation frontiers. We conclude that the main challenges faced by large carnivore conservation in the Amazon are deforestation associated with increasingly frequent and more severe anthropogenic fires. Using a snapshot of threat factors, we also provide a shortlist of protected areas that deserve immediate conservation attention for jaguars and all co-occurring forest biodiversity.

The future of jaguars, even in the most intact Neotropical regions, such as the Amazon and the Pantanal wetlands, is only secure in protected areas where land-use restrictions can be strictly enforced and relentless political pressure to downsize, downgrade and degazette PAs can be resisted. *De facto* law enforcement, as opposed to protection "on paper" only, of healthy Amazonian ecosystems and their apex predators, will therefore require much greater political commitment and investments than we have recently witnessed. Further, considering that this has been established as the decade of ecological restoration by the United Nations (UN) (https://www.decadeonrestoration.org/), our results reinforces the notion that global societies must strive to effectively promote restoration within and between key PAs, as mandated by the UN and signatory nation-states. Despite global frameworks, like CMS, the Jaguar 2030 Roadmap, and the UN Restoration Decade, there is often a divide between high-level commitments and on-the-ground realities. Our results help bridge this gap by prioritizing areas that need action and highlight the key role of forest reserves for jaguar conservation, raising pressure on Brazil as a convention signatory to scale up conservation implementation, moving away from the narrative that simply "holding ground" and stemming losses, represents a conservation triumph.

## Methods

**Study meta-region: Brazilian Amazon**. The Brazilian Amazon represents ~50% of Brazil's territory and ≦76.8% (i.e., 5.15 million km²) of the ~7.59 million km² Pan-Amazon, spanning nine South American countries. The Amazon region contains the vast majority of all Brazilian protected areas. However, the network of protected areas across the Brazilian Amazon is under increasing pressure linked to deforestation and other illegal activities[75,76]. This vast biome is characterized mainly by tropical moist broadleaf forests[77] and has a human population within Brazil of ~23 million people, 72% of which living in major cities[78]. We scoped this study to include all officially sanctioned protected areas across the Brazilian Amazon, including 117 conservation units and 330 indigenous reserves, amounting

to 1,755,637 km$^2$ (Supplementary Figure 1) which represents 41.7% of Brazilian Amazonia, summed by an additional buffer zone of 484,453 km$^2$. Among these 117 conservation units, 57 are strictly protected areas (SPAs) and 60 are sustainable-use reserves (SURs). Among the 330 indigenous reserves, 37 have only been 'declared' as such (grouped as IR type IR1; lands for which the Declaratory Ordinance by the Minister of Justice was issued and are authorized to be physically demarcated, including the delimitation of landmarks and georeferencing), whereas 293 have already been physically demarcated and officially sanctioned (grouped as IR type IR2). These are physically demarcated and georeferenced indigenous territories, and later ratified by a Presidential decree and/or lands that, after the ratification decree, were registered in the Notary Office in the name of the Union and in the Heritage Secretary of the Union) (Supplementary Data 1).

**Data acquisition.** We used several high-resolution spatial layers (rasters or polygons) to extract variables for each protected area that represent (1) a proxy of direct jaguar mortality (e.g., roadkills and persecution due to livestock depletion[53]: (i) human population density (HPD), sourced from the Brazilian Institute of Geography and Statistics (spatial scale of 1:250,000)[79]; (ii) road density (including paved and unpaved roads), sourced from the Brazilian Institute of Geography and Statistics (spatial scale of 1:250,000)[80]; (iii) pasture area (also considered as habitat degradation), sourced from MapBiomas (v.5, spatial resolution of 30 m)[81]; and layers that represent (2) habitat loss and degradation: (i) fire hotspots over a 5-year period (2016–2020), sourced from the National Institute for Space Research-INPE (TERRA satellite MODIS sensor; 1 km spatial resolution)[82]; (ii) deforestation over 4 years (2016–2019) sourced from PRODES (30-m spatial resolution)[82]; and (iii) mining areas, sourced from MapBiomas (v.5, 30 m spatial resolution)[81]. We also obtained the size of each protected area and its adjacent 5 km buffer zone.

Based on the SIRGAS-2000 UTM-ZONE 22°S projection, spatial data extraction was performed separately for both the internal PA area and the external 5 km buffer based on the administrative polygons of each protected area. We used a conservative 5 km buffer threshold because this is approximately the minimum radius for the home range of Amazonian jaguars (i.e., 4.7 km for females, *ca.* 79 km$^2$ considering a radial buffer)[12,19]. Given that the average is a radius of 6.7 km[12,19], the conservative 5 km buffer represents an additional area of 448,452.72 km$^2$ (8.7% of Brazilian Amazon). We sourced data on jaguar population density inside each protected area from ref. [25]. Data extraction was conducted using the ArcGIS 10.8 software[83] based on the average or sum of pixels/area both inside and outside each PA, independently of spatial overlap (pixel *vs.* PAs) area. Further, we obtained the type of legal denomination of each protected area (according to Sistema Nacional de Unidades de Conservação (SNUC[65]), based on Ministério do Meio Ambiente (MMA[84]), and the stage of legal implementation of each indigenous reserve (i.e., declared, approved, physically demarcated and legally sanctioned) sourced from Fundação Nacional dos Povos Indígenas[85].

**Statistics and reproducibility**
*Jaguar density bias and buffer size evaluation.* We formally evaluated the predicted jaguar population densities across all Amazonian PAs as derived from ref. [25] through: (1) a descriptive exploration of the standard errors (*se*) derived from each pixel inside PAs and their respective 5 km buffer; (2) contrasting the predicted values of jaguar density from ref. [25] with in situ estimates based on field studies across the Neotropics previously compiled by Tobler and Powell[86]; and (3) comparing jaguar densities at 13 sites within and immediately around Brazilian Amazonia—based on published (see ref. [87], ref. [88], ref. [89]—with the values within a 5 km radial buffer at the same geographic coordinates derived from ref. [25]. To further assess buffer sizes, we also extracted the jaguar density estimate within a 10 km buffer, which were then regressed against those within a 5 km buffer.

*Jaguar population responses to threats and threat index (TI) for protected area prioritization criteria.* We tested for differences among PA types (i.e., IR1, IR2, SPA, SUR) in how the main response variable (jaguar population size) responded to our environmental predictors (see below) using ANOVAs followed by Tukey post-hoc comparisons by correcting for data asymmetry using $\log_{10}(x + 1)$[90]. We constructed a "*threat index*" (TI) applied to each of the 447 protected areas using the above geospatial layers for both each PA (*in*) and each respective buffer polygon (*out*), which are weighted according to specialized literature on jaguar threats (see refs. [4,16,23,25,49,91–95]). For instance, the major causes of jaguar declines is a synergistic effect of habitat loss, fragmentation, and killings (generally linked to human population density) (e.g., ref. [23], ref. [91], ref. [92], ref. [93], ref. [94]), therefore, these variables received the largest weight in our TI, whose sum can be larger than 1.0 due to synergetic effect upon mammal populations[93]. Yet, other major causes such as roadkill, mining and wildfires frequency and severity also impact directly jaguars across the tropics[95] but comparatively low—until now—than deforestation[23] and killing[94].

To do so, the TI incorporated the following variables calculated for both the PAs ("inside") and their 5 km buffer areas ("outside"): (1) ratio of mining threats (min), defined as the size of mining operations (km$^2$) in relation to PA size (km$^2$), (2) pasture area (*pas*) defined as the size (km$^2$) of pasture areas both inside and outside PAs, (3) ratio of deforestation area over a 4-year time-series (*def*), based on the amount of cumulative deforestation (km$^2$) in relation to PA size; (4) total length (km) of roads (*roa*) overlapping each PA; (5) density of fires (*fir*) defined as

the fire frequency over the 5-year time-series divided by the PA size; and (6) the maximum human population density (*hpd*) for each polygon area. We thus assigned relative weights to these variables to compose the TI according to literature, weighting the threats inside PAs asymmetrically in comparison with the threats outside (i.e., 0.65 vs. 0.35), given that PAs have irreplaceable roles to retain the biodiversity[96]. We also rescaled the threat index given the maximum value at any protected area, which therefore ranged from 0 to 1 by dividing any $TI_i$ for the max $TI_{i,j}$. The threat index—ranging from 0.0 to 1.0—was obtained given the following equation (Equation 1):

$$TI_{i-protected\ area} = \left( \frac{\begin{array}{l}0.65 \times (\sum min_{in} \times 0.05 + pas_{in} \times 0.25 + def_{in} \times 0.50 + roa_{in} \times 0.10 + fir_{in} \times 0.15 + hpd_{in} \times 0.35) \\ + 0.35 \times (\sum min_{out} \times 0.05 + pas_{out} \times 0.25 + def_{out} \times 0.50 + roa_{out} \times 0.10 + fir_{out} \times 0.15 + hpd_{out} \times 0.35)\end{array}}{max\ TI_{i,j}} \right)$$

(1)

To identify PAs with the highest priority for short-term jaguar conservation action, we constructed a bivariate plot between jaguar population sizes inside any PA vs. the threat index (TI). To obtain the jaguar population sizes inside PAs, we used the more conservative value of jaguar density estimates sourced from ref. [25] (i.e., density average − 1.SE) and then categorizing them into population density classes (i.e., 0.00 = <0.01; 0.01 = 0.01–0.02; 0.02 = 0.02–0.03; and 0.03 = >0.03) to reduce the uncertainty of the estimates. Based on the average of both variables at the bivariate plot, we defined one quadrant of short-time high-priority (ST-HP) based on large conservative jaguar populations confronting high threat indices. We then identified the top-10 PAs for which conservation efforts should be allocated across the Brazilian Amazon, selecting areas located in the extreme distribution across the ST-HP quadrant by adding a tangential line within the quadrant that separates the top-10 areas in terms of largest jaguar population sizes vs. highest TIs. Once we identified the main spatial covariates related to jaguar population sizes, we then tested differences between high- and low-priority PAs using an ANOVA followed by Tukey post-hoc tests while correcting for data asymmetry using $\log_{10}(x + 1)$[90]. All data analyses were performed in R version 4.0.5[97] and the complete dataset and data summary are available as Supplementary Data (Supplementary Data 4 and Supplementary Data 5).

**Reporting summary.** Further information on research design is available in the Nature Portfolio Reporting Summary linked to this article.

## Data availability
The data (including R code[97]) that support the findings of this study are openly available online in the additional files (Supplementary Data 1, Supplementary Data 2, Supplementary Data 3, Supplementary Data 4, and Supplementary Data 5) of this manuscript.

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

## Acknowledgements

We thank the WWF network for the financial support. We sincerely thank the Instituto Pró-Carnivoros (https://procarnivoros.org.br/) for the support. We thank Luísa G. L. das Chagas for her support in spatial data extraction. J.A.B. was supported by the São Paulo Research Foundation (FAPESP) postdoctoral fellowship grants 2018-05970-1 and 2019-11901-5. Fernanda D. Abra (ViaFAUNA: http://www.viafauna.com.br/) kindly provided the jaguar drawing used in the figures. We sincerely thank three anonymous reviewers for their important contributions to this manuscript.

## Author contributions

J.A.B.: conceptualization, supervision, data acquisition, data analysis and figures, writing and revising the original draft; V.B.: funding acquisition, project administration, conceptualization, data acquisition, review and editing; C.A.P.: data acquisition, review and editing; M.E.M.S.C.: data acquisition, review and editing; R.G.M.: conceptualization, review and editing; M.O.d.C.: funding acquisition, project administration, conceptualization, data acquisition, review and editing.

## Competing interests

The authors declare no competing interests.
