## [Peer Review File · Communications Biology]

Reviewers' comments:

Reviewer #1 (Remarks to the Author):

This paper addresses an important topic for jaguar conservation through evaluating the status of and threats to Brazilian Amazonian protected areas and indigenous lands within the context of jaguar conservation. The importance lies in that the Brazilian Amazon is the jaguar's population stronghold and is threatened by multiple anthropogenic pressures. Such an analysis is sorely needed since outside of the Amazon jaguar populations have decreased considerably and a reliable evaluation of the status and future prospects of the Brazilian Amazon for the conservation of the jaguar is a prerequisite for the conservation of the species.

Despite the value and the overall quality of the analysis presented I have some major concerns over some aspects of the analysis. In particular I am very uncomfortable with the analysis being based upon extrapolated jaguar density estimates from Jędrzejewski et al. 2018. Despite the value of Jędrzejewski et al. 2018, there are multiple assumptions and data limitations from that work that generate uncertainty that can be propagated in your analysis. Furthermore, what concerns me is that conclusions based upon such an analysis, without taking into account this uncertainty, can potentially be erroneous and lead to poor decision making.

I would not say that this is a case of garbage in and garbage out, but this analysis is a model of a model which concerns me. The extrapolated density values of Jędrzejewski et al. 2018 are based upon a diversity of density estimates, many of those questionable as they were based upon non-spatial capture-recapture modeling with assumptions about sample areas and some with insufficient sampling grid sizes to generate reliable density estimates. Moreover, Jędrzejewski et al. 2018 included no density estimates for the Brazilian Amazon in their analysis, while their scale of analysis is range-wide rather than the spatial scale of protected areas and reserves. That touches on another issue I had with the analysis which was increasing the resolution to 5 km² by disaggregating the spatial results of Jędrzejewski et al. 2018. This is just producing a false resolution since the scale of inference from Jędrzejewski et al. 2018 is 10 km². If an analysis is based upon results at 10 km² the inference should remain at that scale as a minimum.

That said, I see merit in the analysis and the covariates evaluated. Obviously, it would be preferred if there were reliable estimates of density, but in lieu of that, I really do not see it necessarily necessary to analyze the quality of an area or the threat levels as a function of density. One potential option is instead of using the raw extrapolated densities from Jędrzejewski et al. 2018, those values could be grouped and classified on an ordinal scale of density classes. At least that way, with some broad categories, the effects of the uncertainties in the analysis of Jędrzejewski et al. 2018 could be acknowledged to a certain extent. However, based upon what is known of jaguar ecology, I recommend considering quantifying each site in relation to the sites' covariate values. There is considerable support in the literature for the effects on jaguar occurrence and density of the covariates chosen in the analysis, and it was based upon that support that the covariates were chosen. The authors cite Thompson et al. 2021, Morato et al. 2016, and Morato et al. 2018, which illustrate the importance of environmental and anthropogenic factors on jaguar movements and sight use. Also, amongst others, Espinosa et al. 2018 has shown the negative effect of roads on jaguars, and the authors cite Woodroffe 2000 and Woodroffe and Ginsberg to support the effects of human density and edge effects of protected area. Solely based upon the values of the covariates used, the value of each site for jaguars can be evaluated, as well as their threats. I believe that such an analysis would be incredibly valuable, while avoiding my concern of generating estimates from a model of a model. In conducting such an analysis, I mention that I would be comfortable with comparing site assessments with the extrapolated densities of Jędrzejewski et al. 2018 post-hoc as support of the values of the site assessment, rather than basing the analysis upon those extrapolated values.

Reviewer #2 (Remarks to the Author):

This manuscript is a laudable effort to examine the full suite of protected and managed areas in the Brazilian Amazon, assess status and trends, and consolidate a list of priority areas for conservation action. I fully-embrace that effort, and in my comments to the authors, have expressed conceptual perspectives, and itemized ways in which the paper and the writing therein could be improved. The latter (improving the writing) was no small labor. Following the recommendations that I made should improve the writing considerably. The extent of the minor editorial changes that I have suggested may seem to outnumber and outweigh (swamp) the few conceptual suggestions that I make. However, those conceptual suggestions are: 1) contextualize this work more in relation to the 2030 Jaguar Initiative, the Convention of Migratory Species, and the UN Decade of Restoration. Brazil is engaged with all three, 2) make the selection criteria for the top nineteen areas far clearer; 3) expand map 5A. It currently is useless to a "decision-maker" and if this paper is useless to a decision-maker, then maybe its useless. That is strident, maybe even "out-of-line", but said to emphasize that this paper is intended to be a tool for conservation, so modify it a bit to make it more accessible as such, so it performs well as such, and relate it to multi-lateral environmental conventions and agreements, while linking jaguars to biodiversity/ecosystem services, and climate change mitigation. That's all "packaging", but validly so. There are abundant more explicit observations in the specific comments and recommendations for the authors. Those include the need to clarify how the "top-nineteen priority sites for conservation actions" were selected and to provide a much better map of those sites, their location in Brazil, and their relationships to international borders

Comments for the Authors

First and foremost, this is an important paper and merits publication. It focuses on cutting edge conservation issues. The methods are solid, in their description, and in their practice. It is balanced, serious, sober, and mature. I am honored to have reviewed it.

That said, it still needs some work, specifically the writing. The issue is not so much "what is being said" as "how it's being said". I will go through the paper, and rather exhaustively, suggest ways the expression could be improved. There are quite a few such areas.

At some points my comments may also suggest my thoughts about ways the paper can be improved/where it should go/what it says. I will not distinguish between the two (the how and the what), rather, leaving that up to you.

It's a good paper but if can be made better.

Abstract.

Lines 1 & 2. Great title, great theme.

Line 31, switch in "highest" for "largest". Highest jaguar densities.

Line 33, skip the "We reveal", delete it. Just start with Jaguar, capital (upper case) J.

Lines 36-38, try this alternative. "The predicament of a safe future can only be ensured if protected areas persist and resist downgrading and downsizing due to geopolitical measures"

Introduction.

Line 44, take out "their", switch in "many", However many populations

Line 45, insert "local" vulnerable to local extinction

Line 45, switch out several words. Change "they occur at low densities" to "the species occurs at low densities", switch "has" for have. And switch "requires" for "require". I am not going to type out the improved writing of the entire sentence, but that it will be.

Line 49, delete "clear", and delete "Given that". Both are unnecessary.

Line 50, start right in with "Large-bodied. Upper case "L".

Line 51, period after "ranges" and the two citations, then start a new sentence "These apex predators"

Line 52, switch in "despite" in for "even".

Lines 50 to 53 was too long a sentence. Break it up. This may apply to other parts of the paper where I did not catch it.

Also, sometimes you combine two separate completely distinct elements in one sentence, anywhere

that was done, express them as one sentence.

Line 55, period after "control". Then start new sentence with "They" (have populated the imagination)

Lines 54 to 65 you introduce jaguars as a keystone and emblematic species, but do not do so as an umbrella species – even though 1) you do later on in the paper; and 2) much of this paper is predicated upon jaguars being an umbrella species, and indicator species, for broader biological conservation/biodiversity conservation/ecosystem services etc. Why not mention that (umbrella species) on the front end. If so, use the Thornton et al paper and the Olsoy et al paper as references. Apart from that, why not mention the Jaguar 2030 Roadmap and its objectives, which align with your concerns. It is available via

<https://www.internationaljaguarday.org/jaguar-conservation-roadmap>

Embedded in the broader site's unlikely link of

<https://www.internationaljaguarday.org/#:~:text=Observed annually on November 29,of Central and South America.>

You can access the documents via

<https://www.internationaljaguarday.org/jaguar-conservation-roadmap>

Consider it, since the paper's concerns, and your management objectives coincide with the 2030 Jaguar Conservation Initiative. It may fit into objectives. It should fit into discussion.

Aside from all this, as a scientist, I will add that – the literature on jaguars (specifically) exerting a top-down control, playing a role in trophic cascades, is actually quite limited. It is mostly speculative. There are scant to none papers showing with data the assertion. Just saying. Please be aware of and cognizant of that. Do your best with the best citations that you can find, keep it in, I have no issue with that, but, thinking clearly, there is scant empirical evidence of this mantra that jaguar conservationists like to repeat. Keep it in.

Line 70-71, Delete the entire sentence that starts with "This large..." The sentence is confusing, and I suspect unnecessary.

Line 71 and 72, beware confusions between high densities and high numbers, and do so throughout the paper. They are not the same. High densities, such as 6-8/100km² in Belize, ~ 7/100km² in Hato Piñero in Venezuela, 4.4/100km² in Mathias Tobler's study area – are not the same as high numbers. There is an area x density equation that results in high numbers. It may work even with low densities. In fact, we do not really know what the average density for Amazon terra firme forest in Brazil is – because – likely for reasons of access, very few studies have been conducted there. There is a wealth of data from Andes-Amazon, basically the foot of the Andes in Upper Amazon. There are numbers from Varzea in Mamiraua. But not much from Brazilian terra firme forest. Just sayin'. Your used of Wlod's 1.55/100km² is OK. No issue.

Yet, on line 72 you say high jaguar densities, when I think you mean high jaguar numbers.

Line 72-73 suggested rewrite is as follows: "The Amazon forest still holds high jaguar numbers and 67% of the total jaguar range (~9 million km²), and is the area where the jaguar has the highest probability of survival".

In this case it's a low-density x a giant area computation that results in high numbers.

Roadmap estimates 7 million km² in JCU and Corridors, review it and consider using that (and citing it).

Line 77, switch "driven" for "powered".

Line 78, insert "actions" after "conservation priority-setting"

Actions are what you want, correct? There is a ton of research and talk, no true? What we want is actions, so tailor this paper throughout to assist, support, and motivate actions.

Line 88, between "of" and "Brazilian Amazon" insert a "the. 23% of the Brazilian Amazon".

Line 89, "ostensibly protected by 424 indigenous reserves", change it from "protected on paper" to "ostensibly protected by". Later in paper the key role that indigenous lands can play in jaguar conservation is recognized and handled well. However, here, in the intro, it seems a bit cavalier and light, considering that in some area's "nature" is in better shape in indigenous areas than outside, and in some, it's the boundaries of indigenous areas that represent where conservation is working, more so than protected areas. Unite the front and back of the paper and be a little more serious about indigenous territories here.

Line 97, insert an "A", "A similar study found."

Lines 100-104, another long sentence. Needs to be broken up and can be rephrased.

Line 103, on: "However, even Africa's PA network may continue to deteriorate. Its effectiveness depends upon legal status, and particularly, management actions."

"Across the Neotropics, jaguar population declines also coincide with similar human-induced pressures as lions in Africa."

You like the word "reveal". Use it less.

Line 106, delete "Given that". Start the sentence with "PAs".

Line 107, insert "yet" between biodiversity and they.

Line 108, insert "often as" between are and highly degraded.

Overall – Introduction is good.

Results.

Line 129, "Understory" not "Understory".

Discussion.

General Comments:

Returning to something mentioned in my feedback to the introduction, you may want to seriously consider including mention of the Jaguar 2030 Roadmap. Brazil endorsed it.

Also consider mentioning, as in seriously consider mentioning, the inclusion of jaguars in the Convention of Migratory Species (CMS). Brazil is a signatory. Take a moment to review the commitments that Appendix 1 imply – and consider them, because I suspect your northern protected areas lay close to, if not on, an international border.

And also consider reviewing the UN Decade on Restoration. We are now two years, or 20% into that decade, yet you describe an unrelenting pressure of degradation. The Decade is about halting degradation and accomplishing restoration. Perhaps somewhere in this paper it would be worthwhile to mention how while it would be beautiful to accomplish restoration, but just halting unrelenting degradation – on its own – would be a win.

There is sometimes a chasm between the conversations at the high-level meetings of the multi-lateral conventions and treaties – so why not: 1) use the paper as a way of linking reality to high-level conservation-speak; 2) making clear the implications of inclusion, and endorsements, and signatory status – for on the ground advancements on the ground. In Honduras, it would be sweet to just slow the rapidly hemorrhaging level of habitat loss, and by all appearances, in Brazil as well.

Conservation is guilty of fads. For example, re-wilding, as cool as all that is, and needed and valid – can be-taken to an extreme. You present one many (very many) situations around the globe where simply "holding ground" and stemming losses, can represent a victory.

Also, be sure to link jaguars to biodiversity, to ecosystem services, to climate change mitigation. You already do some. Keep that in there. It broadens to audience and amplifies the relevance.

Bottom line – contextualize this – link it to obligations and conservation tools.

A side note and perhaps out of context, is that I was surprised to see Mamiraua in the top nineteen. I do not consider that site to be fire-prone, I suspect it is not experiencing mass conversions to pasture, it has nearly no roads at all, and human population density, the little that I have been there, seemed low. Perhaps I missed something, but it struck me as puzzling. Did the outlandishly high densities from repeated solid SECR CR CT sampling in there somehow skew things? Was Mamiraua afforded a different density than the rest? If so, consider the proportion of those sampling areas (where D is quite high) to the overall size of the reserve. What are the limits of inference?

Another possibility is that your model uses the absence of certain threats (certainly a pile-up of parameters) to influence yours (this paper's) estimation of density/population size, and thus in some way, that way to assess jaguar population (density and / or size) is in some way confounded with threats, (low level of certain threats affects the population parameters?), albeit perhaps inversely. These calculations have an important role in determining membership on the top nineteen list. Somehow, how this works needs to be clearer (in the analysis and its presentation). I have nothing against Mamiraua, great place, great people. Perhaps/perhaps already more secure than many other

places? It has been highlighted, because for me, as a reviewer, knowing a little about the place, it serves as a "road test" of the modelling, to see how it works, and to raise questions, if there are any. There are some.

Specific items.

Line 193, switch "created" in for "induced".

Lines 194-195, I suggest the following:

Line 194, the network of PAs across the Brazilian Amazon, which is important for jaguar

Line 195, conservation across the species' range, is still fulfilling its role.

Line 202, take out "an overwhelming" and just say "The Brazilian Amazon experienced a multi-faceted spike in"

Line 2018, consider taking out "double whammy" that, being slang, may not be universally understandable in English. Suggested alternative "Cattle ranches in the Amazon have two important negative impacts on jaguars."

Line 2019, represents an opportunity to insert the broader, less jaguar-centric effects of conversion to pasture; while switching out "from" and replacing with "in, consider the following; "in habitat loss for forest wildlife and severe impacts on biodiversity" (and add citations accordingly).

Line 223, switch in "show" where currently it says "showed".

Line 230, delete "As", start with "Jaguars".

Line 230-231, "have large spatial requirements and home ranges, so population density depends..." that means inserting an "and" and deleting an "and" and inserting a "so". It will read better.

Line 232, insert "embedded" between areas and highly-forested.

Line 233, delete the "also". Not necessary.

Lines 243-247, another four-line sentence needing to be segmented.

Line 245, end the sentence at "areas". Start the next one with "this". It will read better.

Conceptual note for this section is that there have been an increasing number of pilot projects (in jaguar range) testing underpasses (and in most cases writing it up). Without encouraging roads where neither needed or well suited to the locale, if they become unavoidable and unstoppable, at least there is this mitigation tool. Can that be quickly and succinctly entered – with citations?

Line 295, delete the word "both".

Lines 296-297, switch "usually loses" where it currently reads "always fares much worse".

Line 303, switch in "mix" where it currently says "potpourri". Switch in "challenges" where it currently says "troubles".

Line 325, switch in "has existed" where, currently it says "persists"?

Conclusions.

Line 345, "drivers" not "drives". Switch in "threatening" for where it currently says "confronting with".

Line 359, consider adding inserting "and investments", after commitment and before than we have recently witnessed.

Methods.

Lines 367-368, delete "tropical" before biome. Biome alone should suffice. Anyways, further in the same sentence it says "tropical broadleaf forest". It would not say that if it were not tropical.

Line 377, insert "already" between have and been, so it reads, have already been.

Line 378, insert "These are lands" (that have their boundaries materialized...)

Line 401-403. That is a tiny home range. It seems a pregnant, or post-partum female. It is small. Keep it as is. Just saying.

Methods generally read solid.

Perhaps the how selected the top twenty areas could be clarified a bit. I have some questions.

Table 1.

They all read either "High TI & high jaguar density" or "High TI and high jaguar pop.size" except Caitutu, which reads "both". Both of what?

I am having a bit of trouble with labelling Mamiraua as High TI. The place is a Shangri la of security compared to many places in Mesoamerica, or better said, all of Mesoamerica, much of the Gran

Chaco, and certainly the arc of deforestation farther south. Maybe I am missing something. Continuing with this thread, maybe high jaguar densities and LARGE jaguar population size (may be better "large"), outweighs TI (?). Does this need to be explained some more? Explored and explained? Both?

Does your model give more weight to jaguar population size and density than to threat indexes? Please clarify.

Are the ways of calculating population size and density – and threat indices – somehow confounded? Explore and articulate.

Perhaps it needs to be clearer – WHY these nineteen are high priorities. I suspect the 19 nearly identical classifications in column 5 of table 1, does not quite capture the essence of their selection.

Lines 417 to 444 cover the population size calculations. OK.

Lines 446 to 462 cover the TI calculations. OK.

Lines 463 to 475, selection of the top 20, I am less sure of what you did, the relative weight of the respective contributing factors. It looks like Mamiraua scored high due to high D and high N, but what about the others, the rest? High D, or High N, or High Threats? I am not clear. It is not clear. That clone-looking column 5 in Table 1 is A) not really informing me; and 2) not really assuring me. Somehow this all needs to be more transparent, or explicit or both, so I and the politicians that you seek to influence feel completely informed.

The spaghetti diagram with the jaguar, has "outside" misspelled in the lower right corner. Pastures do not constitute habitat degradation? Sure, they may lead to direct mortality, but they certainly are degraded habitat too.

I need a better figure for the short-listed nineteen areas. Place, type, and name, on a map, with international borders clear as well. Even if the journal wants you to batch that map with the data (all in one figure), I do not. Its currently difficult to see the where and what – in a paper whose strength will be, or should be – exactly that.

Make the current Figure 5 B, C, D, E one figure, no issue with that.

And make the map, Figure 5A, MUCH larger, and much clearer, ten x clearer. Currently it looks like 9 areas, when you say its 19. Expand that map! I see several areas along the Venezuela, Colombia, and Peru borders. That relates to CMS (of which Brazil is a signatory). Ecuador is a signatory to CMS, as is Peru. There are a ton of hand-wringing papers and articles about conservation challenges. Transcend that genre, and make this paper a tool for conservation – more than it presently is – including through a really good map of the top nineteen.

I am not sure that I have an issue with any of the methods, but somehow – spend sometime making them more explicitly accessible to any and all readers, including the specifics of how, in general, and how for each one, the priority areas – were selected. If one goal of this paper is to have a "real world" impact, these suggested upgrades on the paper may increase its potency.

Reviewer #3 (Remarks to the Author):

The authors explore ways to prioritize different protected areas across the Brazilian Amazon for jaguar conservation. Given the high importance of the Amazon for jaguar conservation, the vast area of the region and limited conservation funding, priority setting is very important and the results from this study are highly relevant.

Unfortunately, the data analysis is flawed, making the results unreliable. The authors used estimated jaguar densities from Jędrzejewski et al. (2018). These densities were estimated using a linear model from published camera trap estimates (actually a combination of a model for density and a model for presence/absence). In the predicted densities, any variation in the original density data is lost (those were the residuals in the Jędrzejewski model). What this means is that the SEM model used in this paper is not really modeling jaguar density so much as modeling the relationship of the predictive variables used in this model against the predictive variables used by Jędrzejewski. Jaguar density is actually removed from this completely. Here an example

$Jag_den \sim PAtype + Pasture + Roads + HPD \dots$ (this paper)

$Jag_den = a + TEMP + NPPmean + NPPsd + NA-SA$ (estimated jaguar density from Jędrzejewski)

This means that:

$a + TEMP + NPPmean + NPPsd + NA-SA \sim PAtype + Pasture + Roads + HPD \dots$ (what is actually being modeled in this paper, no jaguar density in this model)

any relationships between predictive variables and jaguar densities in this paper are actually relationship between predictive variables used by Jędrzejewski and predictive variables used in this study. Since the Jędrzejewski model did not use any anthropogenic threats for the density model there is no way to recover anthropogenic effects on jaguars in this analysis unless those are directly related to environmental variables used in the original model. If the authors would use the same predictive variables that Jędrzejewski used in their model they would get an R^2 close to 1.

Given the above, this paper needs a major revision. The SEM should be completely removed. The threat index will need weights based on some other criteria (literature review, expert knowledge?). Densities from Jędrzejewski could still be used to prioritize PAs (Fig. 4), acknowledging that these are "potential" densities and not true densities (the variance will be underestimated and densities might be higher than true densities, especially in areas where there could be a lot of jaguar killing) but they should not be used as independent variables in any statistical model.

Another thing that I am missing from this paper but that would be really interesting to look at is the effect on climate change on PAs within the jaguar habitat. Several recent climate models predict that large parts of the Amazon might reach a tipping point at which they will transform into savannahs <https://www.nature.com/articles/s41467-020-18728-7>. This would have huge implications for jaguars and the ecosystem as a whole.

L31-32: "... the highest jaguar densities and largest estimated population sizes..."

L35-37: It would be nice to get a bit more information here. How many areas, where are they located.

L37: protected? defended? Instead of proofed.

L37: geopolitical pressures might need some explanation as you have not talked about those in the

abstract. Maybe be a bit more specific?

L48: "... networks of protected areas and connectivity corridors."

L48: "(e.g. 5,6)" looks strange, but check journal formatting guidelines

L48-51: this seems pretty redundant to lines 44-48. Revise first paragraph for redundancies.

L55 "top down control on vertebra populations"

L55-56 I find the combination of top-down control and cultural importance in a single sentence a bit strange.

L61: I would say "Home range sizes tend to increase..." pattern of space use could be understood as home range use (resource selection) and movement patters).

L77: aren't the non-natural fires just a way to clear forest for agriculture? I would list the main drivers here, agriculture, cattle ranching and mining.

L86: should be CU not UC unless you explain that that's the Portuguese abbreviation. Explain how they differ from the PA definition above. Do PAs include all UCs and ILs?

L89: Again explain why you abbreviate indigenous reserves with IL not IR.

L89: "but their fate remains highly uncertain" this statement requires some explanation.

L97-99: "A Similar study was developed across the Paleotropics to evaluate the performance of PAs in maintaining viable populations of African lions (*Panthera leo*) and their prey"

L101: "Despite that, PAs within the lion's range have the potential to..."

L103: "..., even if Africa's PA network will continue to deteriorate. "

L104: ", and the effectiveness depends on their legal status and management" not clear what is meant here. Make a new sentence.

L105-106: "Across the Neotropics the jaguar population declines are driven by human-induced pressures similar to those affecting African lions (see12).

L106-108: So far you have only talked about local factors affecting jaguars (mining, roads, fires). Geopolitical factors refer to international politics and their relations to geographical factors. While this can definitely be important (politics of the Chinese government in Latin America, international commodity marked for wood, beef and soya etc.) this would need a lot more explanations. I would suggest removing this last sentence unless you want to explore these issues in much more detail.

L111: ".. are related to ..." suggests that there is a causal relationship.. that threats are directly affected by populations size... maybe use "... are correlated with..."

L116-118: (3) sounds like a possible circular conclusion in that you might prioritize PAs that face the highest threat. Not sure a hypothesis is needed for this.

L403 Female home range in the cited study was estimated as 130km², 2.53 km is the sigma parameter from the SECR model which has to be multiplied by 2.48 to get home range radius.

L465-467: I would argue that population size is really the most important factor. A small high-density population seems to be of much less value than a large, low-density population. Small high-density populations are limited by available habitat and are likely very isolated.

8th September 2022.

From: Juliano A. Bogoni
Universidade de São Paulo
bogoni@usp.br
Phone: +44 07491 928 133

Reviewer #1

Major comment

- 1) This paper addresses an important topic for jaguar conservation through evaluating the status of and threats to Brazilian Amazonian protected areas and indigenous lands within the context of jaguar conservation. The importance lies in that the Brazilian Amazon is the jaguar's population stronghold and is threatened by multiple anthropogenic pressures. Such an analysis is sorely needed since outside of the Amazon jaguar populations have decreased considerably and a reliable evaluation of the status and future prospects of the Brazilian Amazon for the conservation of the jaguar is a prerequisite for the conservation of the species. Despite the value and the overall quality of the analysis presented I have some major concerns over some aspects of the analysis. In particular I am very uncomfortable with the analysis being based upon extrapolated jaguar density estimates from Jędrzejewski et al. 2018. Despite the value of Jędrzejewski et al. 2018, there are multiple assumptions and data limitations from that work that generate uncertainty that can be propagated in your analysis. Furthermore, what concerns me is that conclusions based upon such an analysis, without taking into account this uncertainty, can potentially be erroneous and lead to poor decision making. I would not say that this is a case of garbage in and garbage out, but this analysis is a model of a model which concerns me. The extrapolated density values of Jędrzejewski et al. 2018 are based upon a diversity of density estimates, many of those questionable as they were based upon non-spatial capture-recapture modeling with assumptions about sample areas and some with insufficient sampling grid sizes to generate reliable density estimates. Moreover, Jędrzejewski et al. 2018 included no density estimates for the Brazilian Amazon in their analysis, while their scale of analysis is range-wide rather than the spatial scale of protected areas and reserves. That touches on another issue I had with the analysis which was increasing the resolution to 5 km² by disaggregating the spatial results of Jędrzejewski et al. 2018. This is just producing a false resolution since the scale of inference from Jędrzejewski et al. 2018 is 10 km². If an analysis is based upon results at 10 km² the inference should remain at that scale as a minimum. That said, I see merit in the analysis and the covariates evaluated. Obviously, it would be preferred if there were reliable estimates of density, but in lieu of that, I really do not see it necessarily necessary to analyze the quality of an area or the threat levels as a function of density. One potential option is instead of using the raw extrapolated densities from Jędrzejewski et al. 2018, those values could be grouped and classified on an ordinal scale of density classes. At least that way, with some broad categories, the effects of the uncertainties in the analysis of Jędrzejewski et al. 2018 could be acknowledged to a certain extent. However, based upon what is known of jaguar ecology, I recommend considering quantifying each site in relation to the sites'

covariate values. There is considerable support in the literature for the effects on jaguar occurrence and density of the covariates chosen in the analysis, and it was based upon that support that the covariates were chosen. The authors cite Thompson et al. 2021, Morato et al. 2016, and Morato et al. 2018, which illustrate the importance of environmental and anthropogenic factors on jaguar movements and sight use. Also, amongst others, Espinosa et al. 2018 has shown the negative effect of roads on jaguars, and the authors cite Woodroffe 2000 and Woodroffe and Ginsberg to support the effects of human density and edge effects of protected area. Solely based upon the values of the covariates used, the value of each site for jaguars can be evaluated, as well as their threats. I believe that such an analysis would be incredibly valuable, while avoiding my concern of generating estimates from a model of a model. In conducting such an analysis, I mention that I would be comfortable with comparing site assessments with the extrapolated densities of Jędrzejewski et al. 2018 post-hoc as support of the values of the site assessment, rather than basing the analysis upon those extrapolated values.

Authors' response: We would like to thank Reviewer #1 for the very positive comments on our manuscript. We believe that the body of specific comments raised by you and the other two Reviewers enabled us to enhance the manuscript quality. Rev#1's major comment is addressed below. We thus believe that our broader body of changes were able to address all your concerns.

- 1) We used multiple approaches to compare the densities derived from Jędrzejewski et al. 2018 with empirical data from Neotropics and particularly the Brazilian Amazon (including their boundary areas; N = 13 studies). Yet, only a few studies in the Brazilian Amazon estimate jaguar densities. The results of these new approaches enable us to conclude that Jędrzejewski et al. 2018 estimates had generated credible values of jaguar densities. Please see these major changes in lines 145-165, 514-526 and the new Figure 1;
- 2) We used your recommendation to perform the analysis based on categorical data. Therefore, we reveal that the potential bias on jaguar density estimates derived from Jędrzejewski et al. 2018 does not preclude nor invalidate the main results of this manuscript. Please see our major addition in lines 553-563 and 201-204. Please see also Figure 3, especially the contrast between panel A and B.

Reviewer #2

Major comment

- 1) This manuscript is a laudable effort to examine the full suite of protected and managed areas in the Brazilian Amazon, assess status and trends, and consolidate a list of priority areas for conservation action. I fully-embrace that effort, and in my comments to the authors, have expressed conceptual perspectives, and itemized ways in which the paper and the writing therein could be improved. The latter (improving the writing) was no small labor. Following the recommendations that I made should improve the writing considerably. The extent of the minor editorial changes that I have suggested may seem to outnumber and outweigh (swamp) the few conceptual suggestions that I make. However, those conceptual suggestions are: 1) contextualize this work more in relation to the 2030 Jaguar Initiative, the Convention of Migratory Species, and the UN Decade of Restoration. Brazil is engaged with all three, 2) make the selection criteria for the top nineteen areas far clearer; 3) expand map 5A. It currently is useless to a “decision-maker” and if this paper is useless to a decision-maker, then maybe its useless. That is strident, maybe even “out-of-line”, but said to emphasize that this paper is intended to be a tool for conservation, so modify it a bit to make it more accessible as such, so it performs well as such, and relate it to multi-lateral environmental conventions and agreements, while linking jaguars to biodiversity/ecosystem services, and climate change mitigation. That’s all “packaging”, but validly so. There are abundant more explicit observations in the specific comments and recommendations for the authors. Those include the need to clarify how the “top-nineteen priority sites for conservation actions” were selected and to provide a much better map of those sites, their location in Brazil, and their relationships to international borders. First and foremost, this is an important paper and merits publication. It focuses on cutting edge conservation issues. The methods are solid, in their description, and in their practice. It is balanced, serious, sober, and mature. I am honored to have reviewed it. That said, it still needs some work, specifically the writing. The issue is not so much “what is being said” as “how it’s being said”. I will go through the paper, and rather exhaustively, suggest ways the expression could be improved. There are quite a few such areas. At some points my comments may also suggest my thoughts about ways the paper can be improved/where it should go/what it says. I will not distinguish between the two (the how and the what), rather, leaving that up to you. It’s a good paper but if can be made better.

Authors’ response: We would like to thank Reviewer #2 for the very positive comments on our manuscript. We are also very grateful for his/her meticulous effort in editing the manuscript and enhancing the expression of the narrative. We believe that the body of specific comments raised by Rev#2 (N = 84) and the other two Reviewers enabled us to enhance the manuscript quality. In relation to your major comment, we elaborate below. We thus believe that our broader body of changes were able to address all of your concerns.

- 1) We contextualized (at the Introduction) and discussed our results according to Jaguar Roadmap 2030, the Convention of Migratory Species, and the UN Decade of Restoration. Please see our response to your specific comments on these issues below;
- 2) We clarified the prioritization criteria (i.e. high threat index vs. high jaguar population size). We also redid Figure 5 (now renamed Figure 6);
- 3) See all our responses to your specific comments below.

Specific comments

- 1) Lines 1 & 2. Great title, great theme.

Authors' response: Thank you!

- 2) Line 31, switch in “highest” for “largest”. Highest jaguar densities.

Authors' response: We agree. Done! Please, see line 31.

- 3) Line 33, skip the “We reveal”, delete it. Just start with Jaguar, capital (upper case) J.

Authors' response: We agree. We changed as recommended. See line 33.

- 4) Lines 36-38, try this alternative. “The predicament of a safe future can only be ensured if protected areas persist and resist downgrading and downsizing due to geopolitical measures”

Authors' response: We agree. We changed as recommended. Lines 36-39.

- 5) Line 44, take out “their”, switch in “many”, However many populations.

Authors' response: We agree. Done! See line 46.

- 6) Line 45, insert “local” vulnerable to local extinction

Authors' response: We agree. Done! Line 47.

- 7) Line 45, switch out several words. Change “they occur at low densities” to “the species occurs at low densities”, switch “has” for have. And switch “requires” for “require”. I am not going to type out the improved writing of the entire sentence, but that it will be.

Authors' response: We agree. We changed as recommended, as follows: (lines 46-48)

- 8) Line 49, delete “clear”, and delete “Given that”. Both are unnecessary.

Authors' response: We agree. Done!

- 9) Line 50, start right in with “Large-bodied. Upper case “L”.

Authors' response: We agree. Done!

- 10) Line 51, period after “ranges” and the two citations, then start a new sentence “These apex predators”

Authors' response: We agree. Done!

- 11) Line 52, switch in “despite” in for “even”.

Authors' response: We agree. Done!

- 12) Lines 50 to 53 was too long a sentence. Break it up. This may apply to other parts of the paper where I did not catch it. Also, sometimes you combine two separate completely distinct elements in one sentence, anywhere that was done, express them as one sentence.

Authors' response: We agree. We changed the sentence as recommended (Lines 51-56). We also have taken the time to go over the entire manuscript again to clarify critical sentences.

- 13) Line 55, period after “control”. Then start new sentence with “They” (have populated the imagination)

Authors’ response: We agree. Done!

- 14) Lines 54 to 65 you introduce jaguars as a keystone and emblematic species, but do not do so as an umbrella species – even though 1) you do later on in the paper; and 2) much of this paper is predicated upon jaguars being an umbrella species, and indicator species, for broader biological conservation/biodiversity conservation/ecosystem services etc. Why not mention that (umbrella species) on the front end. If so, use the Thornton et al paper and the Olsoy et al paper as references.

Authors’ response: We appreciate and agree. We changed the sentence according to your recommendation, including the references (see lines 60-61).

- 15) Apart from that, why not mention the Jaguar 2030 Roadmap and its objectives, which align with your concerns. It is available via <https://www.internationaljaguarday.org/jaguar-conservation-roadmap>. Embedded in the broader site’s unlikely link of <https://www.internationaljaguarday.org/#:~:text=Observed> annually on November 29, of Central and South America. You can access the documents via <https://www.internationaljaguarday.org/jaguar-conservation-roadmap>.

Authors’ response: We appreciate these points and agree. We therefore included this information. Please, see lines 60-66 (Below):

“The jaguar is therefore considered an emblematic flagship and a keystone species^{1,14}. Due to their large spatial requirements, jaguars are also considered an umbrella species^{21,85} and are valuable in conservation planning, ensuring that many other co-occurring species and high-quality habitats are protected²¹. For instance, the Jaguar 2030 Roadmap, a range-wide plan to conserve jaguars in priority landscapes and corridors would additionally benefit a suite of co-occurring vertebrates⁸⁶.”

- 16) Consider it, since the paper’s concerns, and your management objectives coincide with the 2030 Jaguar Conservation Initiative. It may fit into objectives. It should fit into discussion.

Authors’ response: We agree. See our response to your comment #15.

- 17) Aside from all this, as a scientist, I will add that – the literature on jaguars (specifically) exerting a top-down control, playing a role in trophic cascades, is actually quite limited. It is mostly speculative. There are scant to none papers showing with data the assertion. Just saying. Please be aware of and cognizant of that. Do your best with the best citations that you can find, keep it in, I have no issue with that, but, thinking clearly, there is scant empirical evidence of this mantra that jaguar conservationists like to repeat. Keep it in.

Authors’ response: We appreciated and agree. We are totally sympathetic with this concern.

- 18) Line 70-71, Delete the entire sentence that starts with “This large...” The sentence is confusing, and I suspect unnecessary.

Authors’ response: We agree. Done!

- 19) Line 71 and 72, beware confusions between high densities and high numbers, and do so throughout the paper. They are not the same. High densities, such as 6-8/100km² in Belize, ~ 7/100km² in Hato Piñero in Venezuela, 4.4/100km² in Mathias Tobler’s study area – are not the same as high numbers. There is an area x density equation that results in

high numbers. It may work even with low densities. In fact, we do not really know what the average density for Amazon terra firme forest in Brazil is – because – likely for reasons of access, very few studies have been conducted there. There is a wealth of data from Andes-Amazon, basically the foot of the Andes in Upper Amazon. There are numbers from Varzea in Mamiraua. But not much from Brazilian terra firme forest. Just sayin’. Your used of Wlod’s 1.55/100km² is OK. No issue.

Authors’ response: We agree. Following the recommendation of all Reviewers, we included a section in the manuscript to accommodate the comparison across available jaguar population densities in light of the available data (generally very scarce). Please, see lines 145-165 and 513-524. Please also see the new Figure 1. Moreover, we re-extracted the density data considering any pixels overlapping with PAs and their buffers, attaining as much as 2.03 ind/100km² without changes in results from this analysis.

20) Yet, on line 72 you say high jaguar densities, when I think you mean high jaguar numbers.

Authors’ response: We agree. We thus changed “density” to “number”, according your comment #21.

21) Line 72-73 suggested rewrite is as follows:”The Amazon forest still holds high jaguar numbers and 67% of the total jaguar range (~9 million km²), and is the area where the jaguar has the highest probability of survival”.

Authors’ response: We agree. We changed as recommended.

22) In this case it’s a low-density x a giant area computation that results in high numbers.

Authors’ response: We agree. See our response to your comment #20 and #21.

23) Roadmap estimates 7 million km² in JCU and Corridors, review it and consider using that (and citing it).

Authors’ response: We agree and included. Please see lines 83-84.

24) Line 77, switch “driven” for “powered”.

Authors’ response: We agree. Done!

25) Line 78, insert “actions” after “conservation priority-setting”

Authors’ response: We agree. Done!

26) Actions are what you want, correct? There is a ton of research and talk, no true? What we want is actions, so tailor this paper throughout to assist, support, and motivate actions.

Authors’ response: We agree. See our response to your comment #25. We also have taken the time to go over the entire manuscript again to make this issue explicit.

27) Line 88, between “of” and “Brazilian Amazon” insert a “the. 23% of the Brazilian Amazon”.

Authors’ response: We agree. Done!

28) Line 89, “ostensibly protected by 424 indigenous reserves”, change it from “protected on paper” to “ostensibly protected by”. Later in paper the key role that indigenous lands can play in jaguar conservation is recognized and handled well. However, here, in the intro, it seems a bit cavalier and light, considering that in some area’s “nature” is in better shape in indigenous areas than outside, and in some, it’s the boundaries of indigenous areas that represent where conservation is working, more so than

protected areas. Unite the front and back of the paper and be a little more serious about indigenous territories here.

Authors' response: We appreciate this comment. We have made these changes and have taken the time to revise the importance of indigenous reserves.

29) Line 97, insert an “A”, “A similar study found.”

Authors' response: We agree. Done!

30) Lines 100-104, another long sentence. Needs to be broken up and can be rephrased.

Authors' response: We agree. We changed the sentence. See lines 110-114.

31) Line 103, on: “However, even Africa’s PA network may continue to deteriorate. Its effectiveness depends upon legal status, and particularly, management actions.”

Authors' response: We agree. Done!

32) “Across the Neotropics, jaguar population declines also coincide with similar human-induced pressures as lions in Africa.”

Authors' response: We agree. We changed the sentence as recommended.

33) You like the word “reveal”. Use it less.

Authors' response: We agree. We have now avoided the use of this word throughout the manuscript.

34) Line 106, delete “Given that”. Start the sentence with “PAs”.

Authors' response: We agree. Done!

35) Line 107, insert “yet” between biodiversity and they.

Authors' response: We agree. Done!

36) Line 108, insert “often as” between are and highly degraded.

Authors' response: We agree. Done!

37) Overall – Introduction is good.

Authors' response: Thank you for all comments and recommendations on our Introduction.

38) Line 129, “Understory” not “Understory”.

Authors' response: We agree. Done!

39) Returning to something mentioned in my feedback to the introduction, you may want to seriously consider including mention of the Jaguar 2030 Roadmap. Brazil endorsed it.

Authors' response: We agree. Please see our response to your specific comment #15. Please, see also lines 81-83.

40) Also consider mentioning, as in seriously consider mentioning, the inclusion of jaguars in the Convention of Migratory Species (CMS). Brazil is a signatory. Take a moment to review the commitments that Appendix 1 imply – and consider them, because I suspect your northern protected areas lay close to, if not on, an international border.

Authors' response: We agree. In doing so, we included two new paragraphs to address all issues raised by your comments #40, #41, #42, #43 and #44. Please, see lines 311-319 and 412-421.

- 41) And also consider reviewing the UN Decade on Restoration. We are now two years, or 20% into that decade, yet you describe an unrelenting pressure of degradation. The Decade is about halting degradation and accomplishing restoration. Perhaps somewhere in this paper it would be worthwhile to mention how while it would be beautiful to accomplish restoration, but just halting unrelenting degradation – on its own – would be a win.

Authors' response: We agree. See our response to your comment #40.

- 42) There is sometimes a chasm between the conversations at the high-level meetings of the multi-lateral conventions and treaties – so why not: 1) use the paper as a way of linking reality to high-level conservation-speak; 2) making clear the implications of inclusion, and endorsements, and signatory status – for on the ground advancements on the ground. In Honduras, it would be sweet to just slow the rapidly hemorrhaging level of habitat loss, and by all appearances, in Brazil as well.

Authors' response: We agree. See our response to your comment #40.

- 43) Conservation is guilty of fads. For example, re-wilding, as cool as all that is, and needed and valid – can be-taken to an extreme. You present one many (very many) situations around the globe where simply “holding ground” and stemming losses, can represent a victory.

Authors' response: We agree. See our response to your comment #40.

- 44) Also, be sure to link jaguars to biodiversity, to ecosystem services, to climate change mitigation. You already so some. Keep that in there. It broadens to audience and amplifies the relevance.

Authors' response: We agree. See our response to your comment #40.

- 45) Bottom line – contextualize this – link it to obligations and conservation tools.

Authors' response: We agree. Please see our response to your comment #40.

- 46) A side note and perhaps out of context, is that I was surprised to see Mamiraua in the top nineteen. I do not consider that site to be fire-prone, I suspect it is not experiencing mass conversions to pasture, it has nearly no roads at all, and human population density, the little that I have been there, seemed low. Perhaps I missed something, but it struck me as puzzling. Did the outlandishly high densities from repeated solid SECR CR CT sampling in there somehow skew things? Was Mamiraua afforded a different density than the rest? If so, consider the proportion of those sampling areas (where D is quite high) to the overall size of the reserve. What are the limits of inference?

Authors' response: We appreciated this comment. With the new analysis (i.e. based on the highest TI vs. highest jaguar population size) the Mamirauá Sustainable Development Reserve was no longer retained within the top-10. We also believe that the body of minor changes to explain the prioritization criteria now explicitly address these issues.

- 47) Another possibility is that your model uses the absence of certain threats (certainly a pile-up of parameters) to influence yours (this paper's) estimation of density/population size, and thus in some way, that way to assess jaguar population (density and / or size) is in some way confounded with threats, (low level of certain threats affects the population parameters?), albeit perhaps inversely. These calculations have an important role in determining membership on the top nineteen list. Somehow, how this works needs to be clearer (in the analysis and its presentation). I have nothing against Mamiraua, great place, great people. Perhaps/perhaps already

more secure than many other places? It has been highlighted, because for me, as a reviewer, knowing a little about the place, it serves as a “road test” of the modelling, to see how it works, and to raise questions, if there are any. There are some.

Authors’ response: We agree. Please see our response to your comment #46. Please, see the new areas in Figure 6A and Table 1. Please also see our new Figure 1.

48) Line 193, switch “created” in for “induced”.

Authors’ response: We agree. Done!

49) Lines 194-195, I suggest the following: Line 194, the network of PAs across the Brazilian Amazon, which is important for jaguar. Line 195, conservation across the species’ range, is still fulfilling its role.

Authors’ response: We agree. Done!

50) Line 202, take out “an overwhelming” and just say “The Brazilian Amazon experienced a multi-faceted spike in”

Authors’ response: We agree. Done!

51) Line 2018, consider taking out “double whammy” that, being slang, may not be universally understandable in English. Suggested alternative “Cattle ranches in the Amazon have two important negative impacts on jaguars.”

Authors’ response: We agree. Done!

52) Line 2019, represents an opportunity to insert the broader, less jaguar-centric effects of conversion to pasture; while switching out “from” and replacing with “in, consider the following; “in habitat loss for forest wildlife and severe impacts on biodiversity” (and add citations accordingly).

Authors’ response: We agree. Done!

53) Line 223, switch in “show” where currently it says “showed.

Authors’ response: We agree. Done!

54) Line 230, delete “As”, start with “Jaguars.

Authors’ response: We agree. Done!

55) Line 230-231, “have large spatial requirements and home ranges, so population density depends...” that means inserting an “and” and deleting an “and” and inserting a “so”. It will read better.

Authors’ response: We agree. Done!

56) Line 232, insert “embedded” between areas and highly-forested.

Authors’ response: We agree. Done!

57) Line 233, delete the “also”. Not necessary.

Authors’ response: We agree. Done!

58) Lines 243-247, another four-line sentence needing to be segmented.

Authors’ response: We agree. Done!

59) Line 245, end the sentence at “areas”. Start the next one with “this”. It will read better.

Authors’ response: We agree. Done!

60) Conceptual note for this section is that there have been an increasing number of pilot projects (in jaguar range) testing underpasses (and in most cases writing it up). Without encouraging roads where neither needed or well suited to the locale, if they become unavoidable and unstoppable, at least there is this mitigation tool. Can that be quickly and succinctly entered – with citations?

Authors' response: We appreciated this comment. We have now included lines 324-325.

61) Line 295, delete the word “both”.

Authors' response: We agree. Done!

62) Lines 296-297, switch “usually loses” where it currently reads “always fares much worse”.

Authors' response: We agree. Done!

63) Line 303, switch in “mix” where it currently says “potpourri”. Switch in “challenges” where it currently says “troubles”.

Authors' response: We agree. Done!

64) Line 325, switch in “has existed” where, currently it says “persists”?

Authors' response: We agree. Done!

65) Line 345, “drivers” not “drives”. Switch in “threatening” for where it currently says “confronting with”.

Authors' response: We agree. Done!

66) Line 359, consider adding inserting “and investments”, after commitment and before than we have recently witnessed.

Authors' response: We agree. Done!

67) Lines 367-368, delete “tropical” before biome. Biome alone should suffice. Anyways, further in the same sentence it says “tropical broadleaf forest”. It would not say that if it were not tropical.

Authors' response: We agree. Done!

68) Line 377, insert “already” between have and been, so it reads, have already been.

Authors' response: We agree. Done!

69) Line 378, insert “These are lands” (that have their boundaries materialized...)

Authors' response: We agree. Done!

70) Line 401-403. That is a tiny home range. It seems a pregnant, or post-partum female. It is small. Keep it as is. Just saying.

Authors' response: We appreciated this comment. According to the recommendation of Reviewer #3 this has been changed (Lines 483-485).

71) Perhaps the how selected the top twenty areas could be clarified a bit. I have some questions. Table 1. They all read either “High TI & high jaguar density” or “High TI and high jaguar pop.size” except Caitutu, which reads “both”. Both of what?

Authors' response: We appreciated this comment. Yet, see our response to your comment #46.

72) I am having a bit of trouble with labelling Mamiraua as High TI. The place is a Shangri la of security compared to many places in Mesoamerica, or better said, all of Mesoamerica, much of the Gran Chaco, and certainly the arc of deforestation farther south. Maybe I am missing something. Continuing with this thread, maybe high jaguar densities and LARGE jaguar population size (may be better “large”), outweighs TI (?). Does this need to be explained some more? Explored and explained? Both?

Authors’ response: We appreciated this comment. However, see our response to your comment #46.

73) Does your model give more weight to jaguar population size and density than to threat indexes? Please clarify.

Authors’ response: We appreciated this comment. However, see our response to your comment #46.

74) Are the ways of calculating population size and density – and threat indices – somehow confounded? Explore and articulate.

Authors’ response: We appreciated this comment. However, see our response to your comments #46 and your major comment #1.

75) Perhaps it needs to be clearer – WHY these nineteen are high priorities. I suspect the 19 nearly identical classifications in column 5 of table 1, does not quite capture the essence of their selection.

Authors’ response: We appreciated this comment. However, see our response to your comments #46 and your major comment #1.

76) Lines 417 to 444 cover the population size calculations. OK.

Authors’ response: We appreciated this comment.

77) Lines 446 to 462 cover the TI calculations. OK.

Authors’ response: We appreciated this comment.

78) Lines 463 to 475, selection of the top 20, I am less sure of what you did, the relative weight of the respective contributing factors. It looks like Mamiraua scored high due to high D and high N, but what about the others, the rest? High D, or High N, or High Threats? I am not clear. It is not clear. That clone-looking column 5 in Table 1 is A) not really informing me; and 2) not really assuring me. Somehow this all needs to be more transparent, or explicit or both, so I and the politicians that you seek to influence feel completely informed.

Authors’ response: We appreciated this comment. However, see our response to your comments #46 and your major comment #1.

79) The spaghetti diagram with the jaguar, has “outside” misspelled in the lower right corner.

Authors’ response: We agree. Done!

80) Pastures do not constitute habitat degradation? Sure, they may lead to direct mortality, but they certainly are degraded habitat too.

Authors’ response: We agree. Pasture has been changed in both threats. Please see the new Figure 3.

81) I need a better figure for the short-listed nineteen areas. Place, type, and name, on a map, with international borders clear as well. Even if the journal wants you to batch that map with the data (all in one figure), I do not. Its currently difficult to see the where and what – in a paper whose strength will be, or should be – exactly that.

Authors' response: We agree. We redid this figure.

82) Make the current Figure 5 B, C, D, E one figure, no issue with that.

Authors' response: We agree. We redid this figure. Due to severe text limitation, we included this part of figure in Figure 6B.

83) And make the map, Figure 5A, MUCH larger, and much clearer, ten x clearer. Currently it looks like 9 areas, when you say its 19. Expand that map! I see several areas along the Venezuela, Colombia, and Peru borders. That relates to CMS (of which Brazil is a signatory). Ecuador is a signatory to CMS, as is Peru. There are a ton of hand-wringing papers and articles about conservation challenges. Transcend that genre, and make this paper a tool for conservation – more than it presently is – including through a really good map of the top nineteen.

Authors' response: We agree. We have completely replotted this figure.

84) I am not sure that I have an issue with any of the methods, but somehow – spend sometime making them more explicitly accessible to any and all readers, including the specifics of how, in general, and how for each one, the priority areas – were selected. If one goal of this paper is to have a “real world” impact, these suggested upgrades on the paper may increase its potency.

Authors' response: We appreciated this comment. We agree with the body of suggestions raised in this first round of revisions, and we are able to address all of these aforementioned issues.

Reviewer #3

Major comment

- 1) The authors explore ways to prioritize different protected areas across the Brazilian Amazon for jaguar conservation. Given the high importance of the Amazon for jaguar conservation, the vast area of the region and limited conservation funding, priority setting is very important and the results from this study are highly relevant. Unfortunately, the data analysis is flawed, making the results unreliable. The authors used estimated jaguar densities from Jędrzejewski et al. (2018). These densities were estimated using a linear model from published camera trap estimates (actually a combination of a model for density and a model for presence/absence). In the predicted densities, any variation in the original density data is lost (those were the residuals in the Jędrzejewski model). What this means is that the SEM model used in this paper is not really modeling jaguar density so much as modeling the relationship of the predictive variables used in this model against the predictive variables used by Jędrzejewski. Jaguar density is actually removed from this completely. Here an example: $Jag_den \sim PAtype + Pasture + Roads + HPD \dots$ (this paper); $Jag_den = a + TEMP + NPPmean + NPPsd + NA-SA$ (estimated jaguar density from Jędrzejewski). This means that: $a + TEMP + NPPmean + NPPsd + NA-SA \sim PAtype + Pasture + Roads + HPD \dots$ (what is actually being modeled in this paper, no jaguar density in this model). Any relationships between predictive variables and jaguar densities in this paper are actually relationship between predictive variables used by Jędrzejewski and predictive variables used in this study. Since the Jędrzejewski model did not use any anthropogenic threats for the density model there is no way to recover anthropogenic effects on jaguars in this analysis unless those are directly related to environmental variables used in the original model. If the authors would use the same predictive variables that Jędrzejewski used in their model they would get an R^2 close to 1. Given the above, this paper needs a major revision. The SEM should be completely removed. The threat index will need weights based on some other criteria (literature review, expert knowledge?). Densities from Jędrzejewski could still be used to prioritize PAs (Fig. 4), acknowledging that these are “potential” densities and not true densities (the variance will be underestimated and densities might be higher than true densities, especially in areas where there could be a lot of jaguar killing) but they should not be used as independent variables in any statistical model. Another thing that I am missing from this paper but that would be really interesting to look at is the effect on climate change on PAs within the jaguar habitat. Several recent climate models predict that large parts of the Amazon might reach a tipping point at which they will transform into savannahs <https://www.nature.com/articles/s41467-020-18728-7>. This would have huge implications for jaguars and the ecosystem as a whole.

Authors' response: We would like to thank Reviewer #3 for the critical but positive comments on our manuscript. We believe that the body of specific comments raised by Rev#3 and the other two Reviewers enabled us to enhance the manuscript quality. In relation to Rev#3's major comment, this has been dealt with below. We thus believe that our broader body of changes were able to address these concerns.

- 4) We used multiple approaches to compare the jaguar population densities derived from Jędrzejewski et al. 2018 with empirical data from Neotropics and particularly the

- Brazilian Amazon (including the boundary areas of these studies; N = 13 studies). Yet, only a few studies estimating jaguar densities are available in the Brazilian Amazon. The results of these new approaches enable us to conclude that the Jędrzejewski et al. 2018 estimates generated credible values of jaguar density, and thereby can be incorporated as a jaguar density proxy for the Brazilian Amazon. In other words, we are modelling estimates of jaguar population sizes (based on density, but also as a function of PA area) and not the climatic-productivity model used by Jędrzejewski et al. 2018. We also used your recommended references to substantiate our results. Please see these major changes in lines 145-165, 514-526 and the new Figure 1;
- 5) Following the recommendation of Reviewer #1, we performed SEM analysis now using categorical data. Therefore, we reveal that the potential bias on jaguar density estimates derived from Jędrzejewski et al. 2018 does not preclude or invalidate the main results of this manuscript. Please see the major addition in lines 553-563 and 201-204. Please also see Figure 3, especially the contrast between panel A and B;
 - 6) The SEM analysis is a pre-requisite to calculate our threat index (i.e. part of the equation weights). Therefore, together with the above, the SEM analysis is crucial background for our approaches.

Specific comments

- 1) L31-32: "... the highest jaguar densities and largest estimated population sizes..."
Authors' response: We agree. Done!
- 2) L35-37: It would be nice to get a bit more information here. How many areas, where are they located.
Authors' response: We agree. Done. Please see lines 35-39.
- 3) L37: protected? defended? Instead of proofed.
Authors' response: We agree. This sentence was changed according to a recommendation from Reviewer #2. Please see lines 39-40.
- 4) L37: geopolitical pressures might need some explanation as you have not talked about those in the abstract. Maybe be a bit more specific?
Authors' response: We agree. We therefore changed to: "geopolitical pressures (e.g. poor implementation of infrastructure and legal weakness)..."
- 5) L48: "... networks of protected areas and connectivity corridors."
Authors' response: We agree. Done!
- 6) L48: "(e.g. 5,6)" looks strange, but check journal formatting guidelines
Authors' response: We agree. We included the references into the main text.
- 7) L48-51: this seems pretty redundant to lines 44-48. Revise first paragraph for redundancies.
Authors' response: We agree. We carried out minor changes to these sentences, please see the entire first paragraph.
- 8) L55 "top down control on vertebra populations"

Authors' response: We agree. Done!

- 9) L55-56 I find the combination of top-down control and cultural importance in a single sentence a bit strange.

Authors' response: We agree. Thus, we rewrote this sentence. See lines 58-65.

- 10) L61: I would say “Home range sizes tend to increase...” pattern of space use could be understood as home range use (resource selection) and movement patterns).

Authors' response: We agree. Done! See line 73.

- 11) L77: aren't the non-natural fires just a way to clear forest for agriculture? I would list the main drivers here, agriculture, cattle ranching and mining.

Authors' response: We agree. Done! Please, see line 92.

- 12) L86: should be CU not UC unless you explain that that's the Portuguese abbreviation. Explain how they differ from the PA definition above. Do PAs include all UCs and ILs?

Authors' response: We agree. Done.

- 13) L89: Again explain why you abbreviate indigenous reserves with IL not IR.

Authors' response: We appreciate this comment. We corrected this typo.

- 14) L89: “but their fate remains highly uncertain” this statement requires some explanation.

Authors' response: We agree. We changed as following (lines 103-106).

- 15) L97-99: “A Similar study was developed across the Paleotropics to evaluate the performance of PAs in maintaining viable populations of African lions (*Panthera leo*) and their prey”

Authors' response: We agree. Done!

- 16) L101: “Despite that, PAs within the lion's range have the potential to...”

Authors' response: We agree. Done!

- 17) L103: “..., even if Africa's PA network will continue to deteriorate.”

Authors' response: We agree. Done!

- 18) L104: “, and the effectiveness depends on their legal status and management” not clear what is meant here. Make a new sentence.

Authors' response: We agree. We rewrote this sentence. Please, see lines 119-128.

- 19) L105-106: “Across the Neotropics the jaguar population declines are driven by human-induced pressures similar to those affecting African lions (see12).

Authors' response: We agree. Done! See lines 125-126.

- 20) L106-108: So far you have only talked about local factors affecting jaguars (mining, roads, fires). Geopolitical factors refer to international politics and their relations to geographical factors. While this can definitely be important (politics of the Chinese government in Latin America, international commodity market for wood, beef and soya etc.) this would need a lot more explanations. I would suggest removing this last sentence unless you want to explore these issues in much more detail.

Authors' response: We appreciated this comment. Yet, this phrasing was conceived to link the overall introduction with our objectives.

21) L111: “.. are related to ...” suggests that there is a causal relationship.. that threats are directly affected by populations size... maybe use “... are correlated with...”

Authors’ response: We agree. We changed to co-related. Yet, our analysis was conceived to depict the relationship between these metrics.

22) L116-118: (3) sounds like a possible circular conclusion in that you might prioritize PAs that face the highest threat. Not sure a hypothesis is needed for this.

Authors’ response: We appreciate this comment. We rephrased as following: “PAs that should be prioritized for jaguar conservation efforts are precisely those confronting the most severe habitat degradation threats, but still safeguarding high jaguar’s population size.” (Lines 139-141).

23) L403 Female home range in the cited study was estimated as 130km², 2.53 km is the sigma parameter from the SECR model which has to be multiplied by 2.48 to get home range radius.

Authors’ response: We agree. We reformulated the sentence as following: “We used a conservative 5-km buffer threshold because this is approximately two-fold the minimum sigma parameter from the SECR model on home range radius of Amazonian jaguars (i.e. 2.53 km for females, ca. 20 km²)²⁴.” (See lines 468-471). Moreover, we performed an additional analysis to predict the relationship between 5-km and 10-km buffer sizes, that were highly related in terms of jaguar estimate density ($R^2_{adj} = 0.97$; $p < 0.001$; Fig 1D).

24) L465-467: I would argue that population size is really the most important factor. A small high-density population seems to be of much less value than a large, low-density population. Small high-density populations are limited by available habitat and are likely very isolated.

Authors’ response: We agree. In this version of the ms, we use only estimates of jaguar population sizes (vs. threat index) to define the top-10 PAs that should be prioritized in the short term. Please see the new Figure 3 and the new Supporting Information File S4.

Best regards,

Juliano A. Bogoni (on behalf of all co-authors)

Wildlife Ecology, Management, and Conservation Lab (LEMaC)
Forest Science Department, ESALQ
University of São Paulo

Reviewers' comments:

Reviewer #1 (Remarks to the Author):

I still believe that evaluating the status and trends of Amazonian protected areas and indigenous reserves is important and valuable, but I am confused by the insistence of the authors to remain with applying the SEM when it is obvious that it is not necessary based upon what is known about jaguar ecology (the role of forest cover, road density, human population density, etc.). Simply categorizing each area based upon the relevant covariates used in the modeling will give an insight into the status and trend for each area. To me the authors are over-thinking their analysis and making it more complicated than is necessary. In the end this revision is more confusing and flawed than the original submission.

The authors say they took into my suggestion to bin density estimates into categories, rather than use the actual estimates, however, they 1) do not discuss how the estimates were binned, and 2) still include the analysis based upon the raw estimates and base their interpretations upon that model. My suggestion was based upon the uncertainty of the density estimates from Jędrzejewski and the related uncertainty, nor does the analysis propagate the error of those density estimates. So, in the end the authors did not really address the issues that I raised. In the newest version of the manuscripts the authors include density estimates from 13 sites in the Amazon but the source is not presented, nor how the data were collected or the quality of the estimates given. Importantly, if density estimates are available for the Amazon why use those of Jędrzejewski at all since none of the data used in that analysis were from the Brazilian Amazon? Additionally confusing is the use of density estimates based upon an allometric body mass equation since it imparts additional uncertainty as it is a model based solely on body mass and is not taking into account the effects of covariates which is the point of the analysis.

The use of the sigma value from SCR modeling from Tobler et al. is erroneous, as sigma is related to the Euclidean distance of space use during the sampling period for that estimate and very small relative to jaguar movements over time (consult estimates for the Amazon from Thompson et al. 2021). Buffer areas need to be considerably larger.

It appears that the authors are using estimates of absolute numbers of jaguars in areas rather than values standardized as density in some of their extrapolations. This can produce highly erroneous results as population size confounds area with covariate effects on populations (figures 4 and 5). Any interpretations of these results are questionable.

The Fig 1, A and B only highlight my concerns over the variations and uncertainty in density estimates, C is actually not significant at the 95% level, and D should be highly correlated since the estimates from a 5km and 10km buffer are based upon the same inputs. I have no idea what Fig 3 is presenting as the caption gives insufficient information, nor, as mentioned above, why you are including the analysis using the raw density estimates? The results of the modeling need to be presented, there is no way to evaluate the model coefficients.

Reviewer #3 (Remarks to the Author):

While the manuscript has overall improved, the issue of the SEM using predicted variables from another model as an independent variable remains. As pointed out in my initial review, the predicted values don't contain any information beyond what was captured by the original covariates. Any relationships found by this analysis can be traced back to correlations with the original covariates. Therefore, the SEM model cannot give us any further insight beyond what was found in Jędrzejewski et al. (2018) and can actually lead to erroneous conclusions. I attach some R code illustrating this

issue with simulated data. The positive correlation between jaguar density and deforestation might be such a case.

It also does not make any sense to analyze "N jaguars inside" and "N jaguars outside" as "N jaguars outside" is just $0.6 * \text{"N jaguars inside"}$ as described on line 512. The regression coefficients will be the same for both, as can be seen in Figure 3. Nothing is gained here. Furthermore, buffer zones can vary greatly across the Amazon, from well protected with similar jaguar densities to highly degraded, so using an arbitrary offset is hard to justify.

Given that the analysis has not changed, my recommendations stay the same.

- 1) Remove the SEM analysis.
- 2) Use results from Jędrzejewski et al. (2018) and other studies to weigh threats.
- 3) Focus on the protected area prioritization and potential future threats.

Hypothesis (2) and (3) can be tested without the SEM.

L26-27: I would add one or two more introductory sentences. Not sure combining top-down control and population decline goes well in one sentence.

L35: with the largest jaguar population sizes...

L37 short to medium-term actions

L38: at boundaries with neighboring countries

L55: densely populated by humans

L83: with "unnatural" do you mean "human caused"

L464: The home range radius based on that reference would be $6.12 (2.53 * 2.45)$ and the home range size would be 120 km^2 . I have no issue with the buffer, but the numbers should be corrected.

L512: This would really depend on the PA. There are many buffer zones in the Amazon that have similar jaguar densities as the core zone, others have high human impacts and the density would be lower.

18th November 2022.

From: Juliano A. Bogoni
Universidade de São Paulo
bogoni@usp.br
Phone: +44 07491 928 133

Reviewer #1

Major comment

- 1) I still believe that evaluating the status and trends of Amazonian protected areas and indigenous reserves is important and valuable, but I am confused by the insistence of the authors to remain with applying the SEM when it is obvious that it is not necessary based upon what is known about jaguar ecology (the role of forest cover, road density, human population density, etc.). Simply categorizing each area based upon the relevant covariates used in the modeling will give an insight into the status and trend for each area. To me the authors are over-thinking their analysis and making it more complicated than is necessary. In the end this revision is more confusing and flawed than the original submission. The authors say they took into my suggestion to bin density estimates into categories, rather than use the actual estimates, however, they 1) do not discuss how the estimates were binned, and 2) still include the analysis based upon the raw estimates and base their interpretations upon that model. My suggestion was based upon the uncertainty of the density estimates from Jędrzejewski and the related uncertainty, nor does the analysis propagate the error of those density estimates. So, in the end the authors did not really address the issues that I raised. In the newest version of the manuscripts the authors include density estimates from 13 sites in the Amazon but the source is not presented, nor how the data were collected or the quality of the estimates given. Importantly, if density estimates are available for the Amazon why use those of Jędrzejewski at all since none of the data used in that analysis were from the Brazilian Amazon? Additionally confusing is the use of density estimates based upon an allometric body mass equation since it imparts additional uncertainty as it is a model based solely on body mass and is not taking into account the effects of covariates which is the point of the analysis. The use of the sigma value from SCR modeling from Tobler et al. is erroneous, as sigma is related to the Euclidean distance of space use during the sampling period for that estimate and very small relative to jaguar movements over time (consult estimates for the Amazon from Thompson et al. 2021). Buffer areas need to be considerably larger. It appears that the authors are using estimates of absolute numbers of jaguars in areas rather than values standardized as density in some of their extrapolations. This can produce highly erroneous results as population size confounds area with covariate effects on populations (figures 4 and 5). Any interpretations of these results are questionable. The Fig 1, A and B only highlight my concerns over the variations and uncertainty in density estimates, C is actually not significant at the 95% level, and D should be highly correlated since the estimates from a 5km and 10km buffer are based upon the same inputs. I have no idea what Fig 3 is presenting as the caption gives insufficient information, nor, as mentioned above, why

you are including the analysis using the raw density estimates? The results of the modeling need to be presented, there is no way to evaluate the model coefficients.

Authors' response: We again would like to thank Reviewer #1 for the comments on our manuscript. We believe that the body of specific comments raised by you and the other two Reviewers during two rounds of revisions enabled us to substantially enhance the quality of the manuscript. Rev#1's major comment is addressed below. We thus believe that our broader body of changes were able to address all of your concerns.

1. We completely removed the SEM analysis from our manuscript. We followed your recommendation using literature data to weight our "threat index" used for PAs prioritization (see lines 505-553);
2. We followed your recommendation using a conservative estimate of density to obtain the number of jaguar inside each PA. Please, see lines 539-553;
3. We included the sources for the Figure 1 data and removed the allometric body mass equation. Please see lines 492-503;
4. Despite adoption a new conservative approach to obtain jaguar population sizes, we further discussed potential bias derived from the Jędrzejewski et al. (2018) estimates. Please see lines 356-364;
5. We removed the use of sigma value from SCR modeling from Tobler et al. Yet, we found — in Morato et al. 2016 and Tompson et al. 2021 (both in the reference lists) — that 4.7-km (i.e. ca. 5-km) is the radius of the smallest home range size estimate for jaguars in the Amazon (the average is 6.7-km). Moreover, the use of this estimate (i.e. outside PAs) was used only for descriptive statistics, not for the prioritization, criteria (see Methods). This buffer size (5-km) distributed across PAs represents an additional buffer zone of 484,453 km²;
6. Some figures were removed, others reframed. Yet Fig.1 was used to indicate some trends in jaguar population density estimates to evaluate the data sourced from Jędrzejewski et al. (2018). Please, also see our response #2, #3 and #4.

Reviewer #3

Major comment

- 1) While the manuscript has overall improved, the issue of the SEM using predicted variables from another model as an independent variable remains. As pointed out in my initial review, the predicted values don't contain any information beyond what was captured by the original covariates. Any relationships found by this analysis can be traced back to correlations with the original covariates. Therefore, the SEM model cannot give us any further insight beyond what was found in Jędrzejewski et al. (2018) and can actually lead to erroneous conclusions. I attach some R code illustrating this issue with simulated data. The positive correlation between jaguar density and deforestation might be such a case. It also does not make any sense to analyze "N jaguars inside" and "N jaguars outside" as "N jaguars outside" is just $0.6 * \text{"N jaguars inside"}$ as described on line 512. The regression coefficients will be the same for both, as can be seen in Figure 3. Nothing is gained here. Furthermore, buffer zones can vary greatly across the Amazon, from well protected with similar jaguar densities to highly degraded, so using an arbitrary offset is hard to justify. Given that the analysis has not changed, my recommendations stay the same. Remove the SEM analysis; 2) Use results from Jędrzejewski et al. (2018) and other studies to weigh threats; 3) Focus on the protected area prioritization and potential future threats.

Authors' response: We again would like to thank Reviewer #3 for the critical but positive comments on our manuscript. We believe that the body of specific comments raised by Rev#3 and the other two Reviewers, during two rounds of revisions, enabled us to enhance the quality of the manuscript. In relation to Rev#3's major comment, this has been dealt with below. We thus believe that our broader body of changes were able to address these concerns.

1. We completely removed the SEM analysis from our manuscript. We followed your recommendation using literature data to weight our "threat index" used for PAs prioritization (see lines 505-553);
2. Consequently, we removed the regression analysis;
3. We used the outside estimates only for descriptive statistics, not for the prioritization, criteria (see Methods);
4. We agree buffer zones definition are a challenge. Yet, we found — in Morato et al. 2016 and Tompson et al. 2021 (both in the reference lists) — that 4.7-km (i.e. ca. 5-km) is the radius of the smallest estimate living-area for the jaguar on Amazon (the average is 6.7-km). Moreover, the use of this estimate (i.e. outside PAs) was used only for descriptive statistics, not for the prioritization, criteria (see Methods). This buffer size (5-km) distributed across PAs represents an additional buffer zone of 484,453 km².

Specific comments

- 1) Hypothesis (2) and (3) can be tested without the SEM.

Authors' response: We agree. We believe that our new approach (without the SEM) now addresses these issues.

- 2) L26-27: I would add one or two more introductory sentences. Not sure combining top-down control and population decline go well in one sentence.

Authors' response: We agree. Done!

- 3) L35: with the largest jaguar population sizes...

Authors' response: Done!

- 4) L37 short to medium-term actions
We rewrite to "short-term action"

- 5) L38: at boundaries with neighboring countries

Authors' response: We agree. Done!

- 6) L55: densely populated by humans

Authors' response: Done!

- 7) L83: with "unnatural" do you mean "human caused"

Authors' response: Yes. We added this information.

- 8) L464: The home range radius based on that reference would be 6.12 (2.53×2.45) and the home range size would be 120 km². I have no issue with the buffer, but the numbers should be corrected.

Authors' response: We agree. We rewrote this sentence. Please see lines 475-489

- 9) L512: This would really depend on the PA. There are many buffer zones in the Amazon that have similar jaguar densities as the core zone, others have high human impacts and the density would be lower.

Authors' response: We agree. Yet, in this new version we used a new approach to obtain and fine-tune the prioritization criteria. Please see lines 505-553.

Best regards,

Juliano A. Bogoni (on behalf of all co-authors)

Wildlife Ecology, Management, and Conservation Lab (LEMaC)

Forest Science Department, ESALQ

University of São Paulo

Final Revision Instructions

To the Author— Please review the editorial comments and requests below and confirm that changes have been made in the manuscript in the right-hand column. **This document must be uploaded** as a related manuscript file.

Please see our final file submission checklist for information about submitting your revised documents.

Files and General Policies	
Main manuscript file must be in Microsoft Word or LaTeX format. LaTeX and Tex article source files must be accompanied by the compiled PDF for reference. The bibliography must be submitted separately (as a .bib file) or contained within the .tex file.	Yes. The final manuscript was uploaded in Microsoft Word (.docx)
Each Figure must be provided as a separate file and must be supplied whole, with all panels included in a single document. Figures should be provided at a minimum resolution of 300 dpi at final size. Figure files must only contain images (please also leave out labels such as “Figure 1” etc). Figure captions must instead be included within the main manuscript file, grouped together at the end of the document.	Yes. All figures were provided in this form.
All figures, tables, and supplementary items must be cited in the manuscript and numbered in the order in which they appear.	Yes. All these items were cited in the manuscript.
Tables in the main manuscript must be provided in an editable format and should be grouped together at the end of the main manuscript file.	Yes. The table was provided in the end of the manuscript.
Please ensure that all equations are supplied in an editable format upon resubmission. Equations must be numbered sequentially.	Yes. Equations are editable.

Please check whether your manuscript contains third-party images, such as figures from the literature, stock photos, clip art or commercial satellite and map data. We strongly discourage the use or adaptation of previously published images, but if this is unavoidable, please request the necessary rights documentation to re-use such material from the relevant copyright holders and return this to us when you submit your revised manuscript. An appropriate permissions statement must be present in the relative figure caption for any third-party images.	There are no third-party images.
Please check that you have not copied any text directly from published work (even your own) without clear attribution, including one or more references. We run a plagiarism detection software and may need to request additional changes if we identify large blocks of identical text.	Ok. Check.
An updated editorial policy checklist that verifies compliance with all required editorial policies must be completed and uploaded with the revised manuscript. All points on the policy checklist must be addressed; if needed, please revise your manuscript in response to these points. https://www.nature.com/documents/nr-editorial-policy-checklist.pdf. Please note that this form is a dynamic ‘smart pdf’ and must therefore be downloaded and completed in Adobe Reader. This file will not open in an internet browser.	Yes. Done.
The reporting summary will be published alongside your manuscript therefore it needs to accurately represent your work. In this case, please take a closer look at the reporting summary and make sure things are completed correctly. If an item does not apply, for example human participants, I need you to check the NA box next to that item. No section should be left blank.	Yes. Done.

Also, please make sure to include your name and date at the top of the document. If you require a new Reporting Summary form, please download it here: https://www.nature.com/documents/nr-reporting-summary.pdf. Please note that this form is a dynamic 'smart pdf' and must therefore be downloaded and completed in Adobe Reader. This file will not open in an internet browser.	
Your paper will be accompanied by a brief editor's summary when it is published on our homepage. Please approve the draft summary below or provide us with a suitably edited version (no more than 250 characters including spaces). The most-at-risk habitats for jaguars in the Brazilian Amazon are identified and anthropogenic factors including deforestation and agriculture are highlighted as root causes of this habitat destruction.	Yes. We agree.
ORCID Communications Biology is committed to improving transparency in authorship. As part of our efforts in this direction, we are now requesting that all authors identified as 'corresponding author' create and link their Open Researcher and Contributor Identifier (ORCID) with their account on the Manuscript Tracking System (MTS) prior to acceptance. ORCID helps the scientific community achieve unambiguous attribution of all scholarly contributions. For more information please visit http://www.springernature.com/orcid.	Yes. All authors' provided their respective ORCID in the affiliations.

For all corresponding authors listed on the manuscript, please follow the instructions in the link below to link your ORCID to your account on our MTS before submitting the final version of the manuscript. If you do not yet have an ORCID you will be able to create one in minutes. https://www.springernature.com/gp/researchers/orcid/orcid-for-nature-research IMPORTANT: All authors identified as ‘corresponding author’ on the manuscript must follow these instructions. Non-corresponding authors do not have to link their ORCIDs but are encouraged to do so. Please note that it will not be possible to add/modify ORCIDs at proof. Thus, if they wish to have their ORCID added to the paper they must also follow the above procedure prior to acceptance. To support ORCID's aims, we only allow a single ORCID identifier to be attached to one account. If you have any issues attaching an ORCID identifier to your MTS account, please contact the Platform Support Helpdesk at http://platformsupport.nature.com/	
We regularly highlight papers published in Communications Biology on the journal’s Twitter account (@CommsBio). If you would like us to mention authors, institutions, or lab groups in these tweets, please provide the relevant twitter handles in the right-hand column.	@bogoni_eco @CarlosPeres_ @ValeriaBoron @biotello @wwfbrasil
We would welcome the submission of material for the ‘Featured Image’ section on the Communications Biology home page. Images should relate to the content of your manuscript but need not be contained within the paper. Photographs and aesthetically interesting images are preferred; diagrams	We do not have a featured image, only manuscript figures.

are generally not used. Suggestions should be uploaded as a Related Manuscript file. Please provide 1200x675-pixel RGB images. You will also need to submit a completed Image License to Publish. Unfortunately, we cannot promise that your suggestions will be used.	
Supplementary information	
Supplementary Information Format and referencing  ● Supplementary Figures, small Tables, and any supplementary text must be provided in a single PDF. Figures and their captions should be presented together.  ○ If you include a title page, please check that the title and author list matches the main manuscript. ● All Supplementary items must be referred to in the manuscript, and items must be mentioned in numerical order. Please do not include general references to “Supplementary Material”; instead refer to specific items. ● Additional files can be provided as Supplementary Data (Excel files, text files, .zip folders), Supplementary Movies, Supplementary Audio, or Supplementary Software (.zip folder) Supplementary Information files will be uploaded with the published article as they are submitted with the final version of your manuscript. Any highlighting or tracked changes should be removed from the file.	The vast majority of supplementary information are excel files. Therefore, we provided as independent files.
Supplementary items must be cited in a consistent format. Names of items in the Supplementary file(s) must match those used in the main manuscript.	We agree. Done.

We recommend using the following naming formats: Supplementary Figure 1, Supplementary Table 1, Supplementary Data 1, Supplementary Note 1, and Supplementary References.	
Large tables and other data types: We strongly recommend depositing these to suitable repositories (such as Figshare, Dryad, or a data type-specific repository if one exists). Otherwise, these must be supplied as Supplementary Data files. Each file must be labelled as Supplementary Data 1, etc.	There are no large tables or data files.
It's mandatory to provide access to the numerical source data for graphs and charts: We strongly recommend depositing these to suitable repositories (such as Figshare, Dryad, or a data type-specific repository if one exists). Otherwise, all source data underlying the graphs and charts presented in the main figures must be uploaded as Supplementary Data (in Excel or text format). Note that only the data used directly for generating the charts needs to be supplied.	The source of data were provided in the figure legends.
For any Supplementary Files such as those mentioned above that are not included your combined PDF (e.g. Supplementary Data, Movies, Audio, Software), please provide a title and description for each file here in the column to the right. For example: File name: Supplementary Data 1 Description: The source data behind the graphs in the paper	File name: Supplementary Figure 1. Description: Spatial distribution of 447 protected areas used to evaluate the threat to Jaguar populations across the Brazilian Amazon. File name: Supplementary Figure 2. Description: Linear relationship between jaguar population size inside PAs (derived from Jędrzejewski et al. 201825 density estimates) and conservative jaguar

communications biology

population size inside PAs (derived from density average – 1.se, and them categorizing in density classes (i.e. 0.00 = <0.01; 0.01 = 0.01-0.02; 0.02 = 0.02-0.03; and 0.03 = >0.03)).

File name: Supplementary Data 1.

Description: Sociopolitical details of 447 protected areas used to evaluate the threat to Jaguar populations across the Brazilian Amazon. Acronyms are: SPA: strictly-protected conservation units; SUR: sustainable use conservation units (SUR); IR1: declared Indigenous Reserves; and IR2: Indigenous Reserves that were delimited, approved, or ratified.

File name: Supplementary Data 2.

Description: ANOVA and post-hoc Tukey results comparing socio-environmental variables between protected area types across the Brazilian Amazon. SPA: strictly-protected conservation units; SUR: sustainable use conservation units (SUR); IR1: declared Indigenous Reserves; and IR2: Indigenous Reserves that were delimited, approved, or ratified.

File name: Supplementary Data 3.

	Description: List of 74 additional protected area codes, names, size (km²), and legal status of high-priority reserves for jaguar conservation across the Brazilian Amazon. Acronyms are SPA: Strictly Protected Conservation Units; IR2: Indigenous Reserves that were delimited, approved, or ratified; and TI: threat index. See the prioritization approach in Fig. 3C.
Title Page	
Please ensure that the author list provided in our manuscript tracking system matches the author list in the main manuscript.	Yes.
Manuscript title Please ensure the title clearly describes the central finding of the paper. We recommend writing the title as a declarative statement of approximately 15 words or fewer. Be sure to include any key species, protein names, or gene names to ensure optimal retrieval of the paper in database searches. The editors recommend the following title: Impending anthropogenic threats and protected area prioritization for jaguars in the Brazilian Amazon	We agree. We thus changed the manuscript title according to editors' recommendation.
Main text	

Format of the main text Please ensure your manuscript includes the following sections, presented in this order: 1. “Introduction”: The background and rationale for the work. The final paragraph should be a brief summary of the major results and conclusions. The results of the current study must only be discussed in this final paragraph. The Introduction should contain no references to figures or tables. Do not include subheadings.2. “Results” or “Results and Discussion”. This should be split into subheaded sections; we recommend 1 subheading per main figure or table. Figures should not be embedded in the text but submitted separately.a. Do not use more than 1 layer of subheadings.b. A “Conclusions” paragraph can be included only if the results and discussion are combined into a single section.3. “Discussion” (optional), without subheadings.4. Methods, which should be split into subheaded sections. Do not use more than 1 layer of subheadings. To improve readability, we recommend that the main text (Introduction, Results and Discussion) be limited to approximately 5000 words or fewer.	The manuscript is in this form.
Statistical reporting Wherever statistics have been derived (e.g. error bars, box plots, statistical significance) the legend needs to provide and define the n number (i.e. the sample size used to derive statistics) as a precise value (not a range), using	All these information were provided.

the wording “n=X biologically independent samples/animals/independent experiments” etc. as applicable.	
Please include exact p-values where possible. We ask that you also include the name of the statistical test and the estimated effect size. If applicable, please also include the confidence interval.	Yes. Done.
Avoid the use of the word “ significant ” unless referring the results of a statistical test.	Yes. We agree.
Display items	
Figure captions/legends Figures must have a title that will appear above the Figure and a legend that will appear below the Figure (see e.g. https://www.nature.com/articles/s42003-020-1059-1/figures/1) The Figure title must describe the Figure as a whole and must not contain reference to specific figure panels. The Figure legend must refer to and describe all panels. Abbreviations, symbols, colors, and shading present in the Figure must be defined. Please write out the symbols/colors in words (blue circles, red dashed line, etc.) within these definitions. All figure panels must be labelled using lower case letters. Please refrain from referring to sections of figures as top/bottom/left/right/, etc.	The figure titles were presented in bold letters . The legends were provided in sequence.
Axis and panel labels will be published as received. We recommend using a sans-serif font such as Arial or Helvetica.	Yes. Done.

Please define the error bars in each Figure and Supplementary Figure where they are used. One statement at the end of each Figure caption is sufficient if the error bars are equivalent throughout the Figure.	Ok.
Tables in the main text Please check that your Tables comply with the following:  ● Do not include shading or colors. All Tables must contain black and white text only. ● Any bold/italic formatting must be either removed or defined clearly in a Table footnote. ● Where Tables contain images, each image should appear in its own cell in the absence of any text. ● All Tables must have a brief title. 	Yes. The table is in this form.
Please pay close attention to our Digital Image Integrity Guidelines. Also ensure that you retain unprocessed data and metadata files after publication, ideally archiving data in perpetuity, as these may be requested during the peer review and production process or after publication if any issues arise.	Ok.
Methods	
Please ensure that all information present in the Reporting Summary is also in the manuscript. This information is usually most appropriate in the Methods section.	In accordance.
We allow unlimited space for Methods. The Methods must contain sufficient detail such that the work could be repeated. It is preferable that all key	In accordance.

methods be included in the main manuscript, rather than in the Supplementary Information. Please avoid use of “as described previously” or similar, and instead detail the specific methods used with appropriate attribution.	
The Methods should include a separate section titled “Statistics and Reproducibility” with general information on how the statistical analyses of the data were conducted, and general information on the reproducibility of experiments (also those lacking statistical analysis), including the sample sizes and number of replicates and how replicates were defined.	In accordance.
Data Policies	
Please add a Data Availability statement. The Data Availability statement must include:  ● Access details for deposited data, including repository name and unique data ID. ● How source data can be obtained. ● A statement that all other data are available from the corresponding author (or other sources, as applicable) on reasonable request. Note that ‘available upon request’ is only appropriate if immediate data access has not been mandated by our policies or by the editors. See here for more information about formatting your Data Availability Statement: http://www.springernature.com/gp/authors/research-data-policy/data-availability-statements/12330880	The data that support the findings of this study are openly available in the additional files of this manuscript.

Communications Biology has a strong preference for all data to be deposited in an approved repository. In some cases, data deposition may be required by the editor. We recommend the following data repositories:  ● GenBank (all DNA sequence data) ● NHGRI-EBI GWAS Catalog (GWAS summary statistics) ● PGS Catalog (polygenic risk scores) ● Gene Expression Omnibus (Microarray or RNA sequencing data) ● Sequence Read Archive (WGS or WES data) ● Protein Data Bank (protein structural data) ● OSF (neuroimaging raw data and EEG/EMG/MEG raw data) ● Neurovault (unthresholded statistical maps, parcellations, and atlases produced by MRI and PET studies) ● Image Data Resource (microscopy data) ● PRIDE (proteomics data) Data types without a specific repository can be deposited in a generalist repository, such as figshare or Dryad. For an up-to-date list of approved repositories, please visit https://www.springernature.com/gp/authors/research-data-policy/repositories/12327124.	The data that support the findings of this study are openly available in the additional files of this manuscript.
Data citation Please cite datasets stored in external repositories in the main reference list. For previously published datasets, we ask authors to cite both the related research articles and the datasets themselves.	All references on data citation were in the main reference list.

For more information on how to cite datasets in submitted manuscripts, please see our data availability statements and data citations policy.	
Code availability Please include a Code Availability statement, indicating whether and how the code can be accessed, including any restrictions to access. In some cases, the editor may require that code be made immediately available. This section should also include information on the versions of any software used, if relevant, and any specific variables or parameters used to generate, test, or process the current dataset. The Code Availability statement must be provided as a separate section after the Data Availability section. Please see our policy on code availability for more information. http://www.nature.com/sdata/for-authors/editorial-and-publishing-policies#code-avail In addition to making the custom code available, please ensure that the version of the code/software described in the paper is deposited in a DOI-minting repository (eg, Zenodo) and that this DOI is also cited in the main Reference list.	The R code of this study are openly available in the additional files of this manuscript.
End Notes	
Please check that your bibliography complies with the following:  Your bibliography should start with the heading “References”. The references must be numbered in the order of appearance in the text, then tables, then figures. 	In accordance.

 ● Any in-text citations to references (e.g. "Gupta et al. show...") should be followed by their corresponding reference citation number from the reference list. ● Manuscript citations must include journal title, article title, volume number, page or article number or DOI, and year of publication. ● No publication can be present more than once in the reference list. ● No footnotes are permitted in the references or elsewhere. Text should be incorporated into the main text, the Methods section, or the Supplementary Information instead. ● Websites should only be listed in the references if they are in common use or curated. ● Where possible, preprints in the reference list should be updated with details of the published, peer-reviewed paper. ● Citations should be formatted in the text using superscript numbers. 	
Please provide a 'Competing interests' statement using one of the following standard sentences:  ● The authors declare the following competing interests: [specify competing interests] ● The authors declare no competing interests. See our competing interests policy for further information: https://www.nature.com/nature-research/editorial-policies/competing-interests	The competing interests were included.
Please provide an 'Author Contributions' section that individually lists the specific contribution of each author to the work. Each author must be referred to by name or initials. Where multiple authors possess identical initials, they must be clearly disambiguated from one another.	Yes. Done.

See our author contributions policy for further information:

<https://www.nature.com/nature-research/editorial-policies/authorship#author-contribution-statements>